# Forniceal deep brain stimulation induces gene expression and splicing changes that promote neurogenesis and plasticity

Amy E Pohodich[1,2], Hari Yalamanchili[2,3], Ayush T Raman[2,4], Ying-Wooi Wan[2,3], Michael Gundry[3], Shuang Hao[2,5], Haijing Jin[2,4], Jianrong Tang[2,5], Zhandong Liu[2,4,5], Huda Y Zoghbi[1,2,3,6]*

[1]Department of Neuroscience, Baylor College of Medicine, Houston, United States; [2]Jan and Dan Duncan Neurological Research Institute, Texas Children's Hospital, Houston, United States; [3]Department of Molecular and Human Genetics, Baylor College of Medicine, Houston, United States; [4]Graduate Program in Quantitative and Computational Biosciences, Baylor College of Medicine, Houston, United States; [5]Section of Neurology, Department of Pediatrics, Baylor College of Medicine, Houston, United States; [6]Howard Hughes Medical Institute, Baylor College of Medicine, Houston, United States

**Abstract** Clinical trials are currently underway to assess the efficacy of forniceal deep brain stimulation (DBS) for improvement of memory in Alzheimer's patients, and forniceal DBS has been shown to improve learning and memory in a mouse model of Rett syndrome (RTT), an intellectual disability disorder caused by loss-of-function mutations in *MECP2*. The mechanism of DBS benefits has been elusive, however, so we assessed changes in gene expression, splice isoforms, DNA methylation, and proteome following acute forniceal DBS in wild-type mice and mice lacking *Mecp2*. We found that DBS upregulates genes involved in synaptic function, cell survival, and neurogenesis and normalized expression of ~25% of the genes altered in *Mecp2*-null mice. Moreover, DBS induced expression of 17–24% of the genes downregulated in other intellectual disability mouse models and in post-mortem human brain tissue from patients with Major Depressive Disorder, suggesting forniceal DBS could benefit individuals with a variety of neuropsychiatric disorders.
DOI: https://doi.org/10.7554/eLife.34031.001

*For correspondence:
hzoghbi@bcm.edu

## Introduction

Deep brain stimulation (DBS) provides electrical stimulation to the brain through implantation of indwelling electrodes in various specific regions, according to the effect desired (*Perlmutter and Mink, 2006*). DBS has proven effective in relieving the symptoms of movement disorders, most notably Parkinson's disease and essential tremor (*Benabid et al., 1991*; *Miocinovic et al., 2013*). More recently, DBS was serendipitously discovered to improve memory in a patient undergoing hypothalamic/fornix stimulation for the treatment of morbid obesity and this further expanded the potential applications of DBS to disorders with impairments in hippocampal memory (*Hamani et al., 2008*). The safety and efficacy of forniceal DBS to activate the hippocampus is now being assessed in human clinical trials as a therapy to slow the cognitive decline resulting from Alzheimer's disease (*Laxton et al., 2010*; *Lozano et al., 2016*; *Ponce et al., 2016*; *Sankar et al., 2015*). Animal studies have shown that forniceal stimulation in mice and rats with defects in hippocampal memory markedly improves deficits in hippocampus-dependent memory tasks; these improvements correlated with enhanced circuit function and neurogenesis (*Hao et al., 2015*; *Shirvalkar et al., 2010*).

**eLife digest** Many brain disorders cause impairments in learning and memory. These include developmental disorders such as Rett syndrome, in which children struggle with learning, as well as diseases of old age such as Alzheimer's disease. People with these disorders often show changes in a region of the brain called the hippocampus. Named after the Greek word for 'seahorse' because of its shape, the hippocampus has a key role in forming new memories. Restoring normal activity in the hippocampus could thus help reduce learning and memory impairments.

One way to increase activity in the hippocampus is through a technique called deep brain stimulation (DBS). As the name suggests, DBS involves lowering electrodes deep into the brain to stimulate specific areas of brain tissue. Applying DBS to the fornix, a bundle of nerve fibers that carries information into and out of the hippocampus, improved memory in a mouse model of Rett syndrome. But exactly how DBS produced this improvement is unclear.

Pohodich et al. now showed that DBS of the fornix alters gene activity in the mouse hippocampus. To activate a gene, cells first used the gene's DNA as a template to produce a molecule of RNA. They then used the RNA as a template to produce a protein. Disorders such as Rett syndrome disrupt the efficiency of this process for large numbers of genes. Pohodich et al. showed that stimulating the fornix of mice with Rett syndrome reversed these changes for about a quarter of genes affected in the disorder. DBS also induced changes in thousands of RNA molecules in healthy mice. Many of these come from genes that support communication between nerve cells or that promote the formation of new nerve cells.

DBS may have beneficial effects in other disorders too. Pohodich et al. showed that DBS also increases the levels of genes that are decreased in two other intellectual disability disorders, as well as genes that are altered in depression. For DBS to become a viable treatment, clinical trials must establish its safety and efficacy. Such trials are already underway in patients with Alzheimer's disease. Future trials could include patients with learning impairments, or with depression that has not responded to other treatments.

DOI: https://doi.org/10.7554/eLife.34031.002

These provocative results led us to explore the possibility of using DBS to treat childhood intellectual disability disorders. We were particularly interested in Rett syndrome (RTT) (OMIM # 312750), a postnatal neurodevelopmental disorder caused by loss-of-function mutation in the X-linked gene methyl-CpG-binding protein 2 (*MECP2*) (*Amir et al., 1999*). RTT was an intriguing candidate for DBS because it involves so many gene expression changes in so many neuronal groups that the chance of developing a viable pharmacotherapy seems exceedingly remote (*Baker et al., 2013*; *Chahrour et al., 2008*; *Johnson et al., 2017*; *Sugino et al., 2014*). We administered forniceal DBS to awake, freely-moving wild-type and *Mecp2*-heterozygous mice (a reliable model of RTT), adjusting stimulation intensities in each mouse to ensure we did not trigger seizures, and found that DBS improved hippocampus-dependent contextual fear and spatial memory behaviors in both wild-type and RTT mice compared to sham-treated mice (*Hao et al., 2015*). In fact, the effect of DBS in the RTT mice was so dramatic that it restored these behaviors to wild-type levels. These improvements correlated with increases in long-term potentiation (a marker of synaptic plasticity) and adult hippocampal neurogenesis, both of which are impaired in RTT mice. In a subsequent study, we found that forniceal DBS restores hippocampal circuit function in *Mecp2*-mutant mice (*Lu et al., 2016*). Determining the mechanism underlying these effects became our next step, and we considered changes in transcription, RNA splicing, and DNA methylation.

Much of neuronal plasticity derives from transcriptional changes triggered by an external stimulus, with the genes that respond earliest to neuronal activation encoding transcription factors that set off a cascade of further alterations (*Ebert and Greenberg, 2013*; *Greer and Greenberg, 2008*). The downstream targets of these transcription factors encode synaptic proteins and signaling molecules that modulate neuronal synaptic properties (*Cohen and Greenberg, 2008*; *Ebert and Greenberg, 2013*). Activity-dependent changes in alternative splicing also contribute to the complexity of the neuronal response, by altering the relative abundance of different transcript variants within the cell, which in turn can alter synaptic physiology (*Iijima et al., 2011*; *Mu et al., 2003*; *Quesnel-*

*Vallières et al., 2016*). Thus, changes in gene expression at both the gene and isoform levels play important roles in shaping the neuronal proteome in response to stimuli.

A growing body of evidence indicates that DNA methylation is also involved in learning and the formation of memories (*Feng et al., 2010*; *Halder et al., 2016*; *Zovkic et al., 2013*). Changes in DNA methylation patterns following activity have been observed in the mouse hippocampus and cortex, and plasticity-associated genes show an enrichment for activity-regulated methylation sites (*Guo et al., 2011*; *Halder et al., 2016*; *Ma et al., 2009*). Stimulus-induced alterations in DNA methylation can be correlated with gene expression changes in these brain regions, as well as with discreet aspects of the learning process (*Guo et al., 2011*; *Halder et al., 2016*). These findings suggest that DNA methylation can be dynamically regulated to influence the behavior of a neuron.

We therefore asked how these processes might be affected by acute forniceal DBS. We first characterized the gene expression and splicing changes induced by activity in dentate gyrus (DG) cells of adult wild-type (WT) male mice. We then ascertained whether changes in the neuronal proteome or the DNA methylome were altered post-DBS and how these changes relate to the transcriptional events observed following stimulation. Next, to evaluate how the response to DBS may differ in diseased neurons, we characterized the transcriptional changes elicited by acute DBS in male mice that lack *Mecp2*. We compared these acute transcriptional changes with those observed in female, *Mecp2*-heterozygous (Het) mice and also assessed the impact of chronic forniceal stimulation on gene expression in Het mice. Finally, we compared acute DBS-induced gene expression changes in wild-type male mice with transcriptional changes observed in other mouse models of intellectual disabilities and in post-mortem tissue from individuals with Major Depressive Disorder.

## Results

### Deep brain stimulation promotes expression of genes involved in neuronal plasticity

To better understand the effects of forniceal DBS on neuronal processes, we utilized a DBS implantation and stimulation paradigm previously reported by our lab and that uses parameters similar to those used in humans (*Figure 1A and B*) (*Hao et al., 2015*). We optimized the paradigm to acutely activate DG neurons and found that 45 min of forniceal DBS with a 20 min recovery period elicited robust activity-dependent gene expression in the DG of WT mice (*Figure 1—figure supplement 1*).

Using this acute activation paradigm, we first performed RNA-sequencing (RNA-Seq) using dentate gyrus tissue from WT mice to obtain an unbiased look at early changes in neuronal transcription post-DBS. We found thousands of gene expression differences between wild-type mice that had undergone DBS treatment and those that had been sham-treated (i.e., implanted with DBS electrodes that did not receive electrical stimulation); 1025 genes were either increased or decreased at least two-fold by DBS (*Figure 1C*; *Figure 1—source data 1*). To see whether DBS induced novel activity-dependent genes or predominantly known activity-dependent genes, we compared genes upregulated at least two-fold by DBS with genes previously reported to have activity-dependent changes in expression in neurons (*Eom et al., 2013*; *Flavell et al., 2008*; *Halder et al., 2016*; *Kim et al., 2010*; *Lin et al., 2008*; *Madabhushi et al., 2015*; *Xiang et al., 2007*). These datasets derived from different brain regions and cell types and employed a variety of activation methods, including drug treatments and physiologic stimuli such as environmental enrichment. Despite these many differences, we found that roughly one-third of the genes increased by DBS in WT mice overlapped with these prior datasets (*Figure 1—figure supplement 1*), and these included many well-characterized immediate early genes such as *Fos* and *Bdnf* in addition to numerous transcriptional regulators and signaling components. Gene ontology (GO) analysis on the genes upregulated by DBS revealed enrichment in signaling components, transcriptional regulators and anti-apoptotic factors (*Figure 1D*; *Figure 1—source data 2*). We validated a number of the gene expression changes we observed in a new cohort of WT DBS mice by RT-qPCR (*Figure 1E*). These data suggest that one means by which DBS influences neuronal behavior is by altering expression of key neuronal genes involved in plasticity.

Whereas the dentate gyrus consists primarily of mature granule neurons, there are other cell types within this tissue that could be activated by DBS and contribute to the gene expression changes. We therefore performed population-specific expression analysis (PSEA), a computational

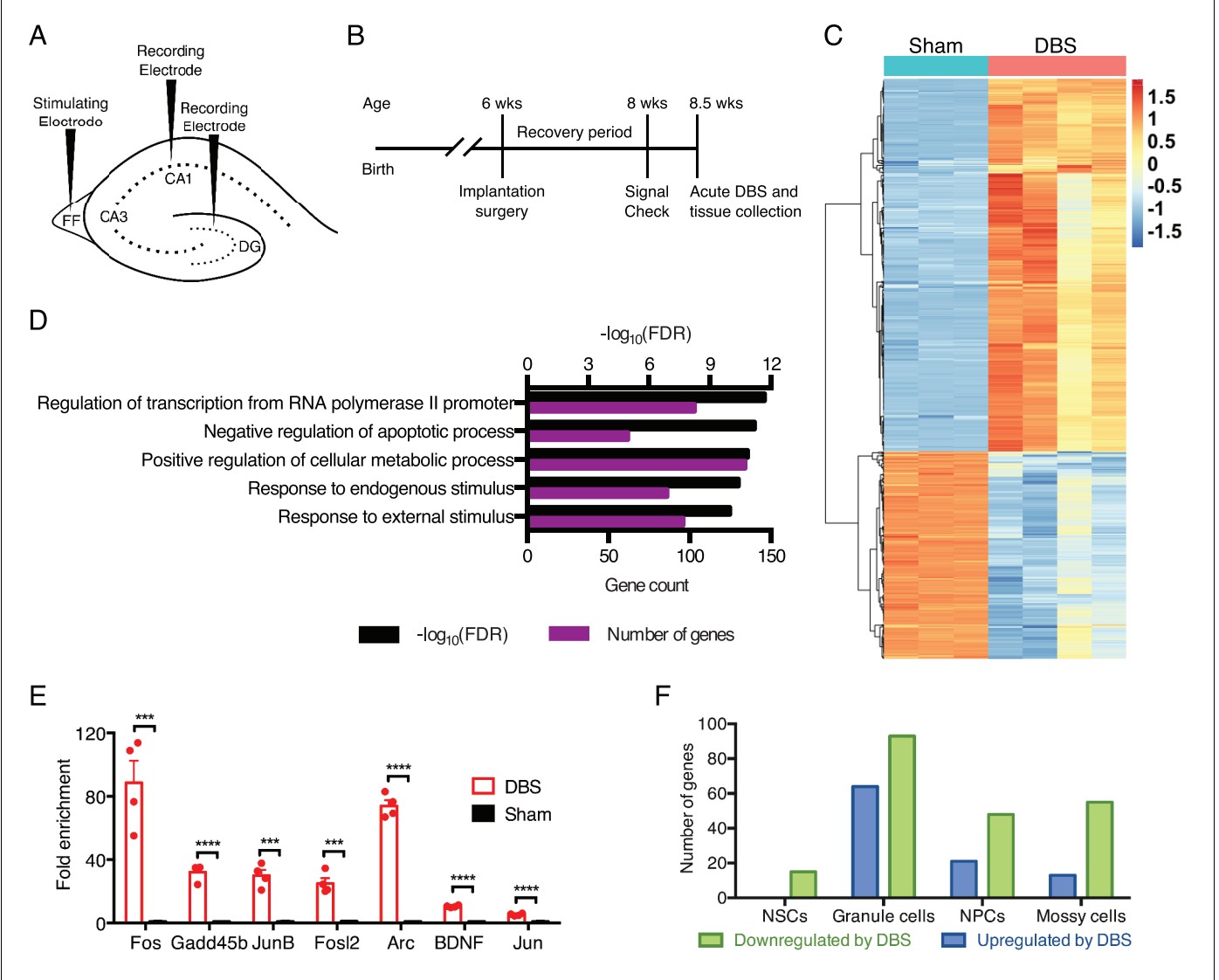

**Figure 1.** DBS alters the expression of many genes, including numerous transcriptional regulators in wild-type mice. (**A**) Schematic showing electrode placement for DBS. DG: dentate gyrus; FF: fimbria-fornix. (**B**) Timeline of implantation and tissue collection used for these studies. Signal check: single pulse induced evoked responses in the DBS pathway as recorded in the hippocampus were verified 1–2 days prior to acute DBS. (**C**) Heatmap showing protein-coding genes whose expression change at least 2-fold (with FDR < 0.05) following DBS in wild-type mice (WT). Sham columns are indicated by teal bar. DBS columns are indicated by the pink bar. (**D**) Gene ontology (GO) analysis of genes that are upregulated at least at least 2-fold (with FDR < 0.05) following DBS. (**E**) RT-qPCR validation of DBS upregulated genes in a new cohort of WT mice (n = 4 sham and 4 DBS mice; significance determined using an unpaired, two-tailed t-test; error bars: SEM; ***p<0.001; ****p<0.0001). (**F**) Numbers of genes found to be significantly altered in expression following DBS in different dentate gyrus cell types (p<0.01). NSCs: Neural stem cells; NPCs: Neural progenitor cells. *Figure 1—figure supplement 1* shows preliminary RT-qPCR experiment data used to pick the optimal duration for acute stimulation, and it shows the overlap of DBS-upregulated genes with prior activity-dependent datasets. *Figure 1—figure supplement 2* shows the stable RNA expression level of genes used as housekeeping genes in RT-qPCR experiments. Source data for all quantified gene expression data in WT sham and DBS samples is provided in *Figure 1—source data 1*. *Figure 1—source data 2* provides the GO data used in *Figure 1D*. The complete PSEA results are available in *Figure 1—source data 3*.

DOI: https://doi.org/10.7554/eLife.34031.003

The following source data and figure supplements are available for figure 1:

**Source data 1.** Quantified gene expression data from wild-type sham and DBS-treated mice.
DOI: https://doi.org/10.7554/eLife.34031.006

**Source data 2.** Gene ontology data for differentially expressed genes in wild-type samples following DBS.
DOI: https://doi.org/10.7554/eLife.34031.007

**Source data 3.** Detailed PSEA results showing cell type and DBS-treatment specific genes.

*Figure 1 continued on next page*

*Figure 1 continued*

DOI: https://doi.org/10.7554/eLife.34031.008

**Figure supplement 1.** Determining stimulation duration for acute DBS and overlap of DBS-induced genes with previously reported activity-dependent genes.

DOI: https://doi.org/10.7554/eLife.34031.004

**Figure supplement 2.** Selection of housekeeping genes for RT-qPCR based on stability of gene expression across genotypes and treatments.

DOI: https://doi.org/10.7554/eLife.34031.005

technique that enables analysis of cell type-specific gene expression in samples comprising heterogeneous cell populations (*Kuhn et al., 2011*). Although many of the genes in our dataset are expressed by multiple cell types, we did find small subsets of genes unique to each cell type assessed (*Figure 1F*; *Figure 1—source data 3*). These findings indicate that DBS likely leads to transcriptional alterations in many dentate gyrus cell types, not just in mature granule neurons.

## DBS induces alternative RNA splicing

RNA splicing changes have been shown to be important for synaptic plasticity and neurodevelopment (*Grabowski and Black, 2001*; *Iijima et al., 2011*; *Mu et al., 2003*), but few studies have had the opportunity and resolution to evaluate how activity affects RNA splicing. We found that DBS caused at least a 30% change in expression of thousands of protein coding isoforms, and a subset of these isoform expression changes occur in genes whose overall expression does not change, indicating possible isoform switches (*Figure 2A*; *Figure 2—source data 1*). GO analysis revealed that these isoforms that are altered with no overall gene-level expression differences are enriched for proteins associated with neurogenesis, morphogenesis, and synaptic function (*Figure 2B*; *Figure 2—source data 2*).

One example of a gene whose overall expression remains the same but whose isoform expression changes is *Kif1b*, a kinesin-family motor protein with an important role in vesicular transport to synapses (*Charalambous et al., 2013*). We found that a shorter isoform of this gene (isoform 201 with transcript ID: NM_008441) was upregulated nearly two-fold upon DBS, despite the lack of change in overall gene-level expression (*Figure 2C*). We were able to validate this isoform expression change in a new cohort of mice by assessing the expression level of the Kif1b isoform 201 by targeting RT-qPCR primers to the unique 3' region of this transcript (unique region of transcript shown in *Figure 2C*, expression level shown in *Figure 2D*). This shorter isoform differs significantly in its protein structure, as it lacks a key pleckstrin homology (PH) domain found in the C-terminus of longer Kif1b isoforms. Because PH domains can bind phosphatidylinositol in cell membranes and interact with protein kinase C and heterotrimeric G proteins (*Wang et al., 1994*; *Yao et al., 1994*), it is possible that this shortened isoform has a localization pattern and signaling component interaction distinct from those of the longer isoforms.

Many of the observed splicing changes involve synaptic and signaling molecules, including the neurotrophic signaling molecule Bdnf and multiple vesicle trafficking proteins in the kinesin family. We were able to validate these changes in a new cohort of mice even when the fold-change in expression of the isoform is small (*Figure 2D*). Notably, in some instances, such as in the case of Bdnf, some of the different isoforms are driven by alternative promoter usage and some by both alternative promoter usage and alternative splicing. Altogether, we find that splicing changes in response to DBS make up a large proportion of the RNA regulatory response in neurons, and many of these changes likely contribute to the improvements in plasticity and neurogenesis observed in mice that underwent chronic forniceal DBS (*Hao et al., 2015*).

## DBS activates jun signaling pathways

Given the numerous splicing and gene expression changes we observed following DBS, we next wanted to determine whether we could identify any early changes in protein levels that may be contributing to the later phases of DBS-induced transcriptional changes. To this end, we performed mass spectrometry analysis to assess whether changes in any key proteins were observed following acute DBS (45 min of forniceal DBS with a 20 min recovery period). Given that these samples were collected at the same time point as those used for RNA-Seq, we did not expect that there would be sufficient time for the levels of many proteins to change drastically. Consistent with this hypothesis,

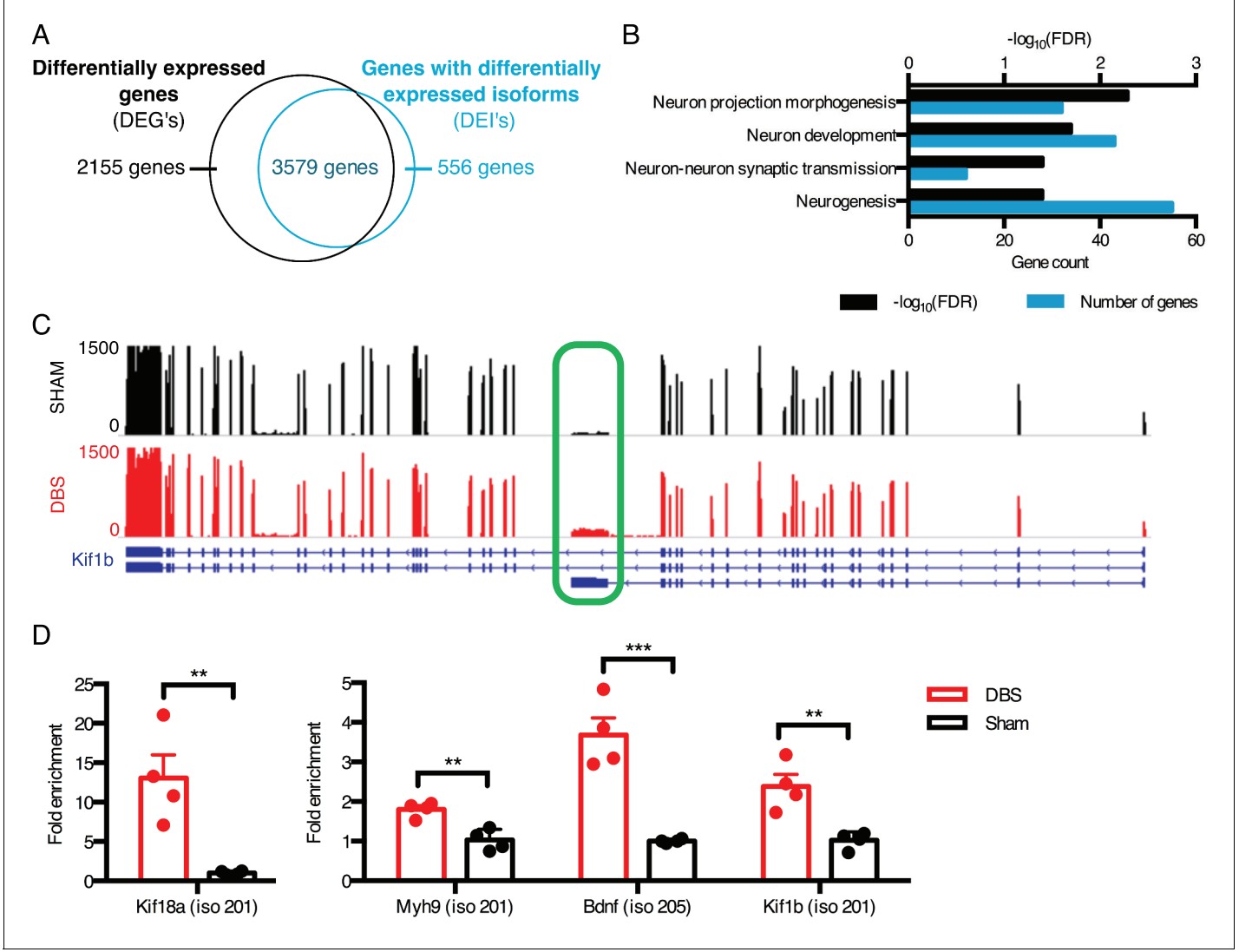

**Figure 2.** DBS revealed hundreds of activity-dependent splicing changes in genes that would be overlooked by differential gene analysis. (**A**) Overlap between genes that are differentially expressed with DBS (fold-change >20%; FDR < 0.05) and genes with differential isoform expression following DBS in WT mice (Fold-change >30%; FDR < 0.05). (**B**) Gene ontology (GO) analysis of genes showing differential isoform expression but not an overall change in gene expression following DBS. (**C**) Representative RNA-sequencing tracks from WT sham (black; max: 1500 reads) and WT DBS (red; max: 1500 reads) mice showing the expression of the *Kif1b* gene, along with annotated Kif1b isoforms (shown in blue). The shortest isoform is differentially expressed post-DBS, and the green box indicates the unique region of the shortest isoform where RT-qPCR primers were located to check transcript levels in a new cohort. (**D**) RT-qPCR validation of DBS upregulated isoforms in a new cohort of WT mice (n = 4 sham, 4 DBS mice; significance determined using an unpaired, two-tailed t-test; error bars: SEM; **p<0.01; ***p<0.001). Source data for RNA isoforms quantification can be found in *Figure 2—source data 1*. The complete list of GO terms and scores for genes with differentially expressed isoforms that are not differentially expressed at the whole gene level can be found in *Figure 2—source data 2*.

DOI: https://doi.org/10.7554/eLife.34031.009

The following source data is available for figure 2:

**Source data 1.** Isoform expression data from wild-type sham and DBS-treated mice.
DOI: https://doi.org/10.7554/eLife.34031.010

**Source data 2.** Gene ontology data for genes in wild-type samples with differentially expressed isoforms (DEIs) that are not differentially expressed at the gene level.
DOI: https://doi.org/10.7554/eLife.34031.011

we found only a few dozen proteins with more than a 50% difference in levels between sham and DBS samples (*Figure 3—source data 1*). Nevertheless, among these differentially expressed proteins we did find three transcription factors that were markedly increased following DBS (*Figure 3A*; *Figure 3—source data 1*).

We performed network analysis using the TRANSFAC database (*Matys et al., 2006*) to identify whether any known targets of these transcription factors were altered in expression in the RNA-Seq

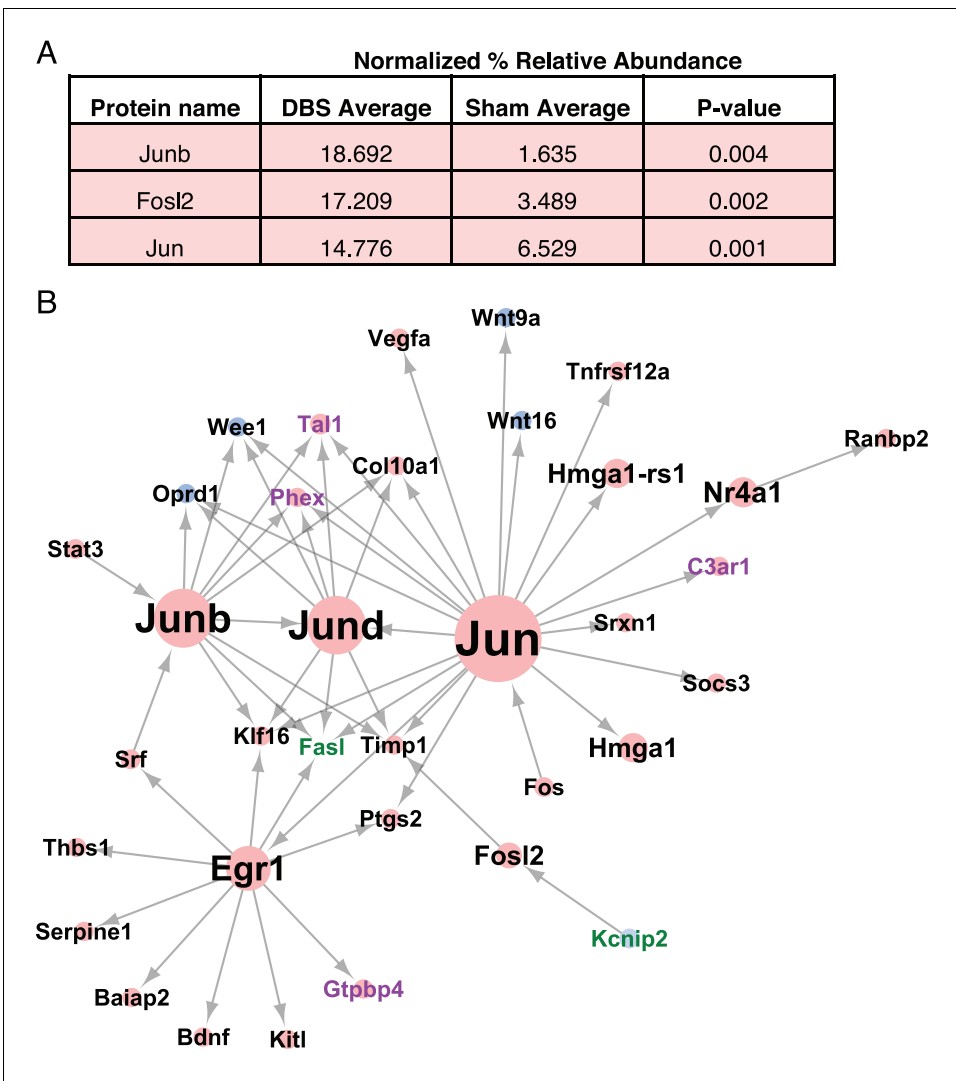

| Normalized % Relative Abundance | | | |
|---|---|---|---|
| Protein name | DBS Average | Sham Average | P-value |
| Junb | 18.692 | 1.635 | 0.004 |
| Fosl2 | 17.209 | 3.489 | 0.002 |
| Jun | 14.776 | 6.529 | 0.001 |

**Figure 3.** DBS induces Jun-associated transcriptional programs. (**A**) Transcription factors found to be increased in WT dentate gyrus neurons following DBS as quantified by Mass spectrometry. P-values were calculated using an unpaired, two-tailed t-test. (**B**) Network analysis results showing the direct and downstream targets of the three transcription factors whose protein levels are altered following DBS. The identified targets are genes that are either significant DEG's (Differentially expressed genes; FDR < 0.05,>2 fold expression change) or DEI's (Differentially expressed isoforms; FDR < 0.05,>2 fold expression change). Node size indicates the number of targeted genes. Node color indicates the log-fold change in expression of that gene, with blue indicating downregulated genes, and red indicating upregulated genes. Label colors indicate the category of gene: Black: the gene is both DEG and DEI, Purple: DEG only, Green: DEI only. The detailed mass spectrometry results from WT sham and DBS mice can be found in *Figure 3—source data 1*.
DOI: https://doi.org/10.7554/eLife.34031.012

The following source data is available for figure 3:

**Source data 1.** Detailed mass spectrometry results for sham and DBS wild-type mice.
DOI: https://doi.org/10.7554/eLife.34031.013

data from the DBS mice. A number of known targets of the Jun family of transcription factors were altered following DBS (*Figure 3B*). When we looked at the targets of the Jun family transcription factors that were present in the DBS dataset, we noticed that some of these direct targets were themselves transcription factors. Many of the transcription factors targeted by the Jun family also targeted genes that were upregulated following DBS, suggesting that the Jun-family signaling cascade may be a key player in DBS-mediated transcriptional changes. It is important to note, however, that while the newly synthesized proteins can contribute to the transcriptional changes we observed, the majority of the gene expression changes seen at the acute time point are likely mediated by pre-existing proteins in the neurons.

## DBS-induced genes have unique DNA methylation patterns

The importance of DNA methylation in the regulation of gene expression has been well-studied during neurodevelopment, and mCG methylation (methylation of a cytosine adjacent to a guanine) has been shown to correlate with a subset of transcriptional changes observed with neuronal activity (*Guo et al., 2011*; *Guo et al., 2014*; *Hon et al., 2014*; *Ma et al., 2009*; *Szulwach et al., 2011*). Little is known, however, about activity-dependent changes in alternatively methylated DNA, specifically at methylated cytosines followed by a nucleotide other than guanine (mCH, where H = A, C or T). Thus, we chose to investigate not only whether DNA methylation changes occurred following DBS, but whether these changes were in mCG or mCH marks, and whether these changes correlated with the gene expression and splicing changes we characterized.

We performed bisulfite sequencing to assess genome-wide methylation patterns in sham- and DBS-treated WT mice and began by evaluating the level of mCG methylation on genes that were increased or decreased by more than two-fold in DBS samples. We found that genes that are upregulated by DBS have lower mCG methylation in their promoters and gene bodies (*Figure 4A*) and this methylation is lower to start with (i.e., in sham-treated mice these genes also show lower mCG levels), suggesting that this lower methylation level is important for the responsiveness of these genes to stimuli. We do not, however, see similarly distinct patterns of mCH methylation in DBS-altered genes (*Figure 4—figure supplement 1*).

We next quantified and compared the mCG and mCH methylation levels of the genome in DBS- and sham-treated WT mice at single base resolution and in 1000 bp bins. Regions with at least a 50% change in mCG or mCH methylation were designated as differentially methylated regions (DMRs), and these DMR's were then assigned functional classifications based on their location, specifically whether they fell in intergenic, intronic, exonic, or promoter regions in the genome (*Figure 4—source data 1*). In comparison with the proportion of the genome falling into one of these four functional categories, DBS-induced base-level changes in mCG methylation were overrepresented in exon and promoter regions and depleted in intergenic regions; changes in mCH methylation patterns followed the genomic distribution (*Figure 4B*; *Figure 4—figure supplement 1*). The finding that mCG methylation changes occur at a higher rate than predicted by the distribution of promoter and exon regions in the genome suggests that these locations may be particularly susceptible to activity-induced changes in DNA methylation, and at least in some instances we found that the DNA methylation change does correlate with gene expression changes. For example, in the gene *Fosl2*, whose protein product changed following acute DBS, we observed that decreases in promoter mCG methylation with activity correlated with increased expression of the gene (*Figure 4C*; promoter region mCG highlighted in yellow boxes). These findings suggest that activity-dependent changes in DNA methylation may contribute to some of the gene expression changes observed post-DBS.

## DBS partially normalizes gene expression in *Mecp2*-deficient mice

In addition to assessing how forniceal DBS affects healthy neurons, we sought to explore how it affected neurons with abnormal physiology. *Mecp2*-heterozygous female mice have impaired hippocampal function and have previously been shown to respond to chronic forniceal DBS (*Asaka et al., 2006*; *Calfa et al., 2015*; *Hao et al., 2015*; *Moretti et al., 2006*; *Weng et al., 2011*). We reasoned that studying DBS-induced gene expression changes in *Mecp2*-deficient mice would afford us the opportunity to evaluate whether the transcriptional response to DBS differs between WT and diseased neurons and gain insight into what changes underlie the benefit these mice derive from

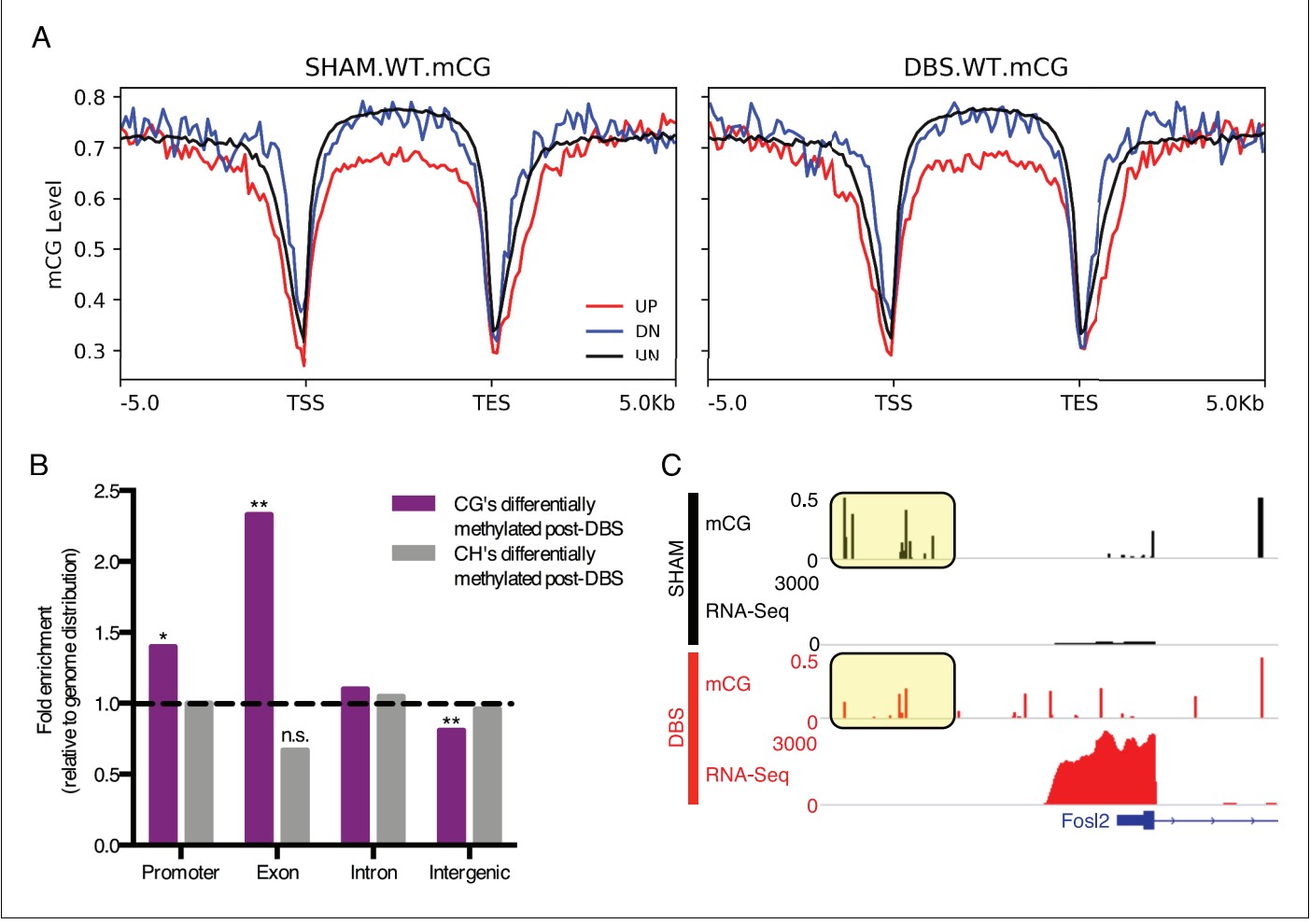

**Figure 4.** DBS-induced transcription and splicing changes show unique methylation patterns. (**A**) Running average plot of mCG methylation levels on DBS upregulated genes (shown in red; FDR < 0.05,>2 fold increase in expression), DBS downregulated genes (shown in blue; FDR < 0.05,>2 fold decrease in expression), and in genes whose expression is unchanged by DBS (shown in black). The left panel shows the mCG methylation of these genes observed in sham samples. The right panel shows the mCG methylation of these categories of genes observed in DBS samples. (**B**) Genomic locations of differentially methylated regions (DMRs) with a greater than 50% change in methylation following DBS. Locations evaluated: intergenic, introns, exons, and promoters. The percentage of DMRs falling into each location type was divided by the percentage of the genome comprised of that location type to generate a fold enrichment score for DMR locations relative to the genomic distribution. Dashed line at y = 1 indicates the genomic values. *p<0.05, **p<0.01. (**C**) *Fosl2* mCG methylation and RNA-sequencing tracks from representative WT sham (shown in black) and WT DBS (shown in red) samples. The annotated gene is shown in blue, and the mCG level in the promoter region is highlighted in the yellow boxes. *Figure 4—figure supplement 1* shows mCH distribution at promoters and gene bodies in DBS-regulated genes, and it also shows genomic localization of DMRs with a greater than 50% change in methylation following DBS. The DMR source data, including p-values, percent change and genomic localization is provided in *Figure 4—source data 1*.
DOI: https://doi.org/10.7554/eLife.34031.014

The following source data and figure supplement are available for figure 4:

**Source data 1.** Locations of differentially methylated regions that are significantly different between wild-type sham and wild-type DBS samples.
DOI: https://doi.org/10.7554/eLife.34031.016

**Figure supplement 1.** mCH levels do not show distinct signatures on DBS up- or downregulated genes.
DOI: https://doi.org/10.7554/eLife.34031.015

chronic forniceal DBS. We used *Mecp2*-null (KO) males for our studies because all of their neurons lack MeCP2 expression, allowing easier detection of gene expression changes.

We first established the baseline gene expression and splicing changes observed in the dentate gyrus of *Mecp2*-null mice (*Figure 5A*; *Figure 5—source datas 1* and *2*). We found that, consistent with other brain regions analyzed in previous studies (*Baker et al., 2013*; *Chahrour et al., 2008*;

*Gabel et al., 2015*), the majority of misregulated genes in KO samples were downregulated (*Figure 5A*). Additionally, we found that roughly equal numbers of genes had altered isoform expression patterns when compared to WT samples (*Figure 5A*). One potential way in which DBS

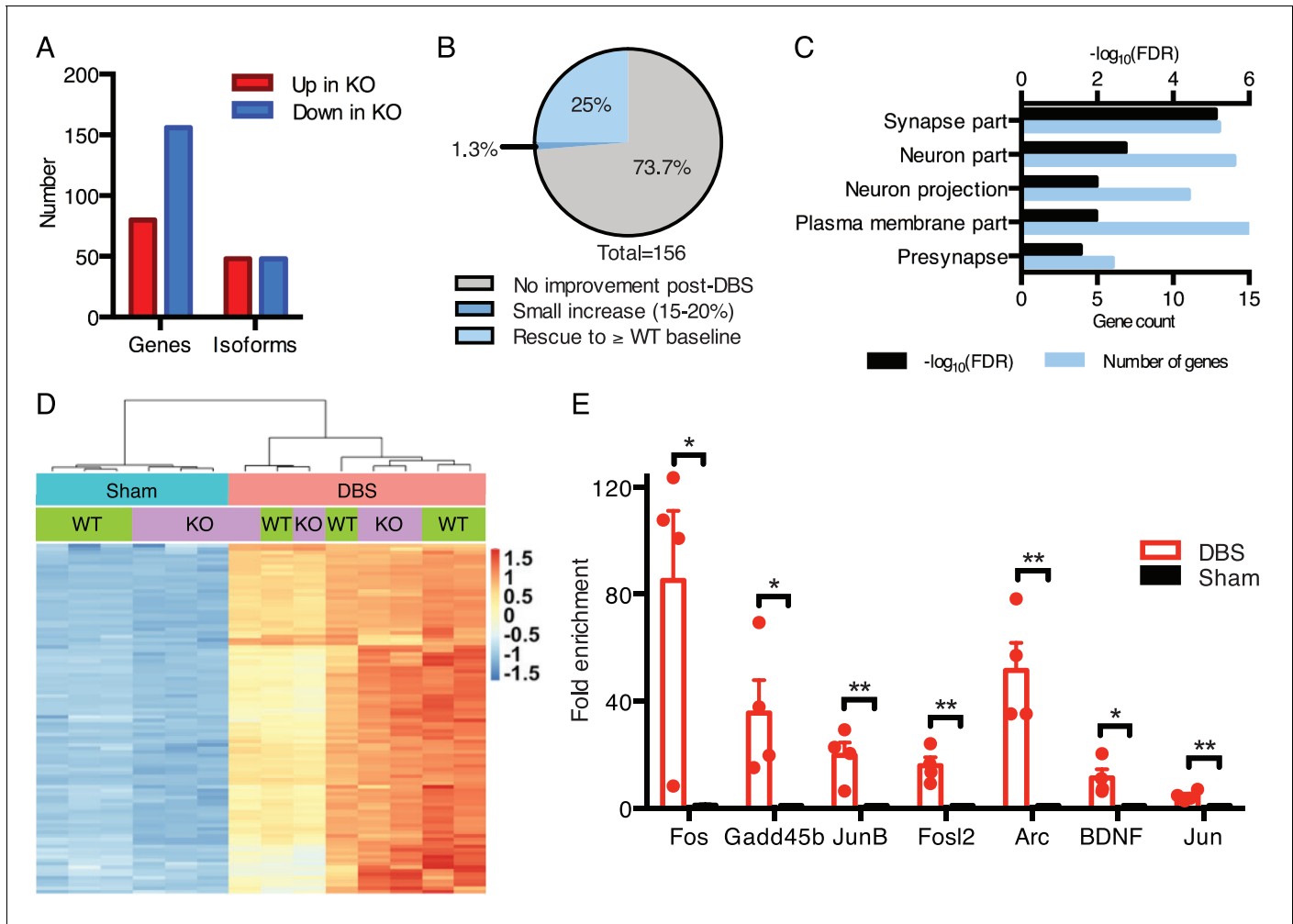

**Figure 5.** DBS rescues expression of genes important for neurological function that are low in *Mecp2*-null mice. (A) Differentially expressed genes and isoforms in KO sham samples (FDR < 0.05,>20% fold-change) as compared to WT samples. (B) Evaluation of the effect of DBS on genes with decreased expression in KO sham samples. Gray: genes with unchanged expression following DBS in KO mice (FDR > 0.05). Dark blue: genes with a small but significant increase following DBS in KO mice (FDR < 0.05; Fold-change <20%). Light blue: genes rescued to at least WT baseline levels (FDR < 0.05, Fold-change >20%). (C) Gene ontology analysis of the genes rescued back to WT baseline levels in KO mice. (D) Heat map showing the expression levels of the top 100 protein coding genes with the highest increase in expression in WT mice, and the comparison in expression of these genes between WT and KO samples. (E) RT-qPCR validation of DBS upregulated isoforms in a new cohort of WT mice (n = 4 sham, 4 DBS mice; significance determined using an unpaired, two-tailed t-test; error bars: SEM; *p<0.05; **p<0.01). All quantified gene expression changes along with comparisons between WT and KO samples at baseline and following DBS can be found in *Figure 5—source data 1*. Isoform expression data from WT and KO mice can be found in *Figure 5—source data 2*. GO data for genes with rescued expression following DBS in KO mice can be found in *Figure 5—source data 3*.

DOI: https://doi.org/10.7554/eLife.34031.017

The following source data is available for figure 5:

**Source data 1.** Quantified gene expression data from wild-type and *Mecp2*-null sham and DBS-treated mice.
DOI: https://doi.org/10.7554/eLife.34031.018
**Source data 2.** Isoform expression data from wild-type and *Mecp2*-null sham and DBS-treated mice.
DOI: https://doi.org/10.7554/eLife.34031.019
**Source data 3.** Gene ontology data for DBS-rescued genes in KO mice.
DOI: https://doi.org/10.7554/eLife.34031.020

could improve the Rett mouse hippocampal phenotype would be to normalize gene expression in the DG. Thus, we next looked at the genes that were low in the KO mice at baseline (i.e., in sham-treated samples) to see if their expression was increased by DBS. We found that 25% (39 genes) of the genes that were downregulated in KO sham-treated mice were restored to at least WT baseline levels following DBS (*Figure 5B*). These restored genes were enriched for important neural functions, including components of synapses and neuronal projections such as *Gad2*, *Grin2d*, and *Syt6*. The rescued expression of these genes likely contributes to the improved plasticity reported with chronic DBS in Rett mice (*Figure 5C*; *Figure 5—source data 3*) (*Hao et al., 2015*).

We next asked how the activity-dependent transcriptional response of MeCP2-deficient mice compared to their wild-type littermates. Despite the notable deficits at baseline, we found that KO mice treated with DBS were capable of upregulating nearly all of the genes we found increased in WT mice post-DBS (*Figure 5—source data 1*). Moreover, the level of induction of these genes was similar between WT and KO mice. To illustrate these findings, we took the top 100 protein-coding genes upregulated by DBS in WT samples, and we performed an unsupervised hierarchical clustering analysis to see how the KO and WT samples compared (*Figure 5D*). We found that samples cluster according to treatment, with sham-treated samples separated from DBS-treated samples, and sham samples were further subdivided based on genotype. However, upon DBS treatment, KO and WT samples did not cluster according to genotype. These results suggest that the mean expression levels of most of these genes are similar between WT and KO samples after DBS. Consistent with this finding, qPCR showed that the same activity-dependent genes assessed in WT DBS samples (see *Figure 1E*) were also significantly upregulated in a new cohort of KO DBS mice (*Figure 5E*).

Given the improvement in gene expression observed in the *Mecp2*-null male mice following acute DBS, we expanded our investigation to determine if similar changes occur following forniceal DBS in WT and *Mecp2*-heterozygous (Het) female mice. We first established that acute stimulation (45 min of DBS with a 20 min recovery period) resulted in the induction of similar activity-dependent genes in female mice (*Figure 6A*). We then assessed the levels of four genes with rescued expression following acute DBS in male KO samples to see if they also increased following acute DBS in female Het samples (*Figure 6B*). Only one gene, Rxfp3, was significantly different between WT and Het sham samples at this early age (female mice were approximately 8.5 weeks of age at time of tissue collection); however, we did observe a significant increase in expression of all four gene in Het mice following acute DBS, similar to changes seen in KO mice. Next, we performed RNA-Seq on samples that received chronic DBS or sham treatment (1 hr of stimulation or sham treatment per day for 14 days followed by a 17 day recovery period prior to tissue collection). This time point was chosen for gene expression analysis because this was the earliest time at which behavior was evaluated and an improvement documented following chronic DBS in Het mice (*Hao et al., 2015*). We found that 63 protein coding genes were significantly increased and 94 protein coding genes were significantly decreased in the Het sham samples compared to WT sham samples (see *Figure 6—source data 1* for a complete list of all quantified gene expression changes; mice were approximately 13.5 weeks of age at the time of tissue collection). We focused on the downregulated genes in Het sham samples to see if their expression was rescued following chronic DBS. We found that 12 of these genes were significantly upregulated (fold-change >15%, FDR < 0.05) following chronic DBS in the Het samples (*Figure 6C*; genes with a significant increase in expression are indicated by asterisks) and that many of the other genes showed a trend toward increased expression following chronic DBS. Further, when we looked at the average expression of genes that are significantly altered in Het sham samples (protein coding genes that are either up- or downregulated by at least 20%, FDR < 0.05), we found that Het DBS samples show a trend toward improved expression of both up- and downregulated genes and that three of the four Het DBS samples even cluster with the WT samples (*Figure 6—figure supplement 1*).

## DBS upregulates genes that are pathologically reduced in other neuropsychiatric diseases

Given the potential for forniceal DBS to improve cognition in Rett patients, we wondered whether it might prove beneficial in other disorders that involve impaired hippocampal function. We searched for gene expression data from the hippocampus of animal models of intellectual disability in the GEO database (*Barrett et al., 2013*). The first dataset we assessed was obtained from *Ptchd1*-KO mice. Loss-of-function mutations in the *PTCHD1* gene lead to X-linked intellectual disability (OMIM #

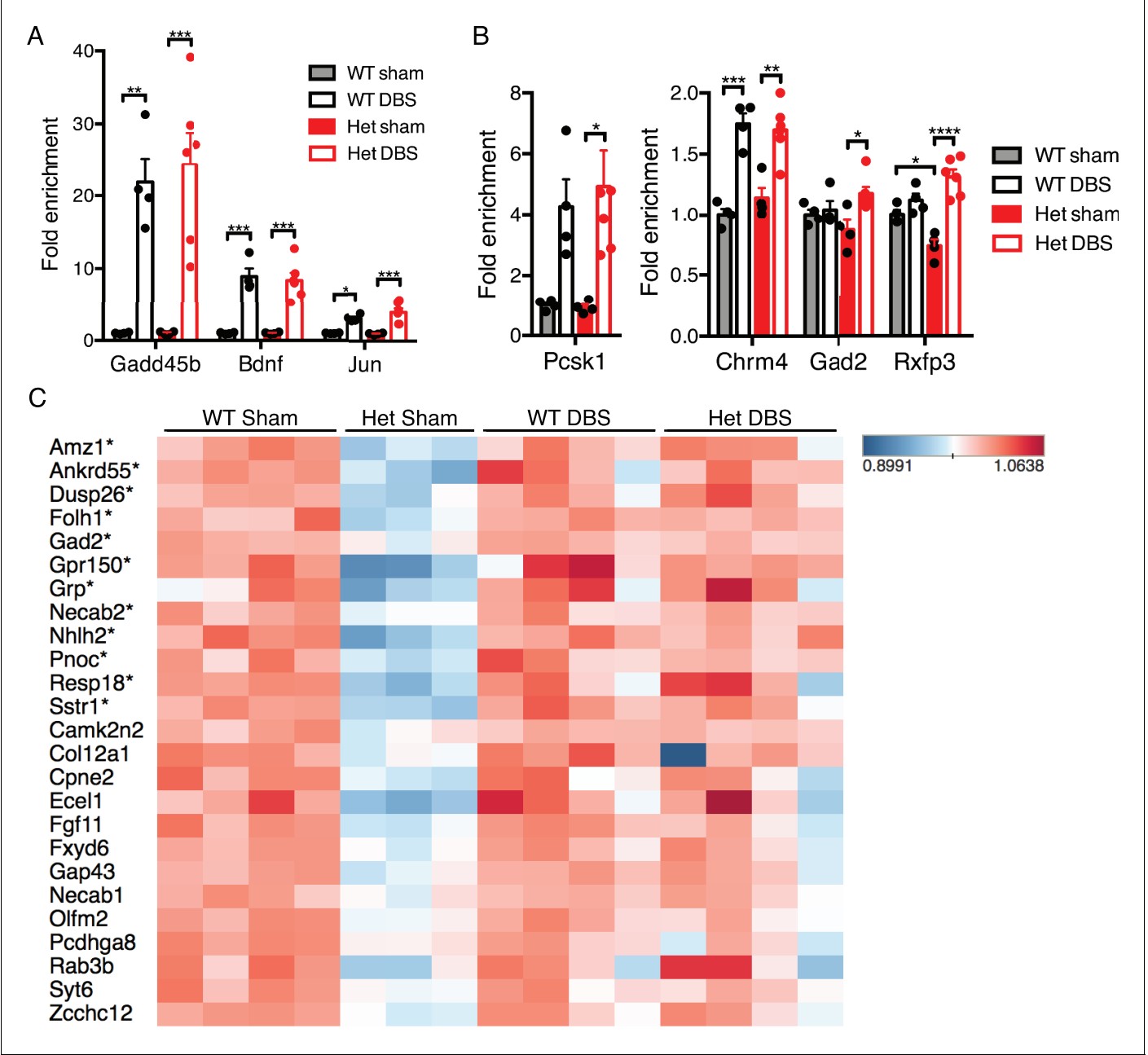

**Figure 6.** Forniceal DBS induces similar acute gene expression changes in *Mecp2*-heterozygous (Het) mice and reveals a trend for sustained gene expression rescue following chronic DBS. (A) and (B) RT-qPCR data from 8.5-week-old female samples following acute DBS (45 min of DBS; n = 4 WT sham, 4 WT DBS, 4 Het sham, and 6 Het DBS; significance determined using an two-way ANOVA with Holm-Sidak multiple comparisons test; error bars: SEM; *$p<0.05$; **$p<0.01$; ***$p<0.001$; ****$p<0.0001$. (C) Heat map of RNA-Sequencing data from 13.5-week-old WT and Het female chronic forniceal DBS samples showing a subset of the protein coding genes that are downregulated by at least 20% (padj $<0.05$) in Het sham samples. Genes with an asterisk (*) are those that are significantly increased (fold-change $>15\%$; FDR $< 0.05$) in the Het samples that received chronic DBS. All quantified gene expression data from chronic DBS and sham treated female mice can be found in *Figure 6—source data 1*. *Figure 6—figure supplement 1* shows a heat map of all genes that are either down- or upregulated at least 20% (FDR $< 0.05$) in Het sham mice.

DOI: https://doi.org/10.7554/eLife.34031.021

The following source data and figure supplement are available for figure 6:

**Source data 1.** Quantified gene expression data from chronic forniceal DBS or sham treated wild-type and *Mecp2*-heterozygous mice.
DOI: https://doi.org/10.7554/eLife.34031.023

**Figure supplement 1.** Heat map showing expression levels of protein coding genes that are significantly altered in *Mecp2*-heterozygous mice.
DOI: https://doi.org/10.7554/eLife.34031.022

300830) (*Ung et al., 2017*). We found that nearly 23% (185 genes) of the genes that are downregulated in the hippocampus of *Ptchd1*-KO mice are amongst the genes that increase following DBS in WT mice (*Figure 7A*). Similarly, a dataset derived from a mouse model of Coffin-Lowry syndrome (OMIM # 303600), a syndromic form of intellectual disability caused by loss-of-function mutations in the *RPS6KA3* gene which encodes RSK2, also showed that nearly 24% (22 genes) of the genes that are downregulated in Rsk2-KO mice are upregulated by forniceal DBS (*Figure 7A*).

We also evaluated a dataset obtained from post-mortem human brain tissue from individuals with major depressive disorder (MDD) (*Duric et al., 2013*). Again, we found a small but significant overlap between this dataset and DBS-upregulated genes, with 17% (325 genes) of the genes that are decreased in expression in the dentate gyrus of patients with MDD being upregulated by forniceal DBS (*Figure 7A*). We were surprised to find that a substantial number of genes in each of these disorders were induced by DBS in WT mice, and that the overlaps between these datasets and DBS-induced genes are significant and not due to chance (Fisher exact test; p<0.001). Further, when we assessed which genes were found in the overlap between these datasets and the DBS data, we saw that few genes were shared among all of the disorders (*Figure 7—figure supplement 1*), indicating that the potential transcriptional benefits of DBS would be different for each disorder.

Finally, given the upregulation of neurogenesis-associated genes following DBS, we also assessed whether a significant overlap could be observed between our DBS dataset and other datasets where neurogenesis is thought to be a key component of the treatment. Specifically, we assessed gene expression data gathered after treatment with either fluoxetine, a selective serotonin reuptake inhibitor (SSRI), or running, both of which have been shown to enhance adult neurogenesis (*Miller et al., 2013*; *Samuels et al., 2014*). When we compared the genes that were upregulated by fluoxetine treatment, we found that approximately 35% (237 genes) are also upregulated by DBS (*Figure 7B*). Nearly 22% (166 genes) of the genes upregulated after 4 days of running are also increased post-DBS (*Figure 7B*). In both comparisons, the overlap with DBS is significant (Fisher exact test;

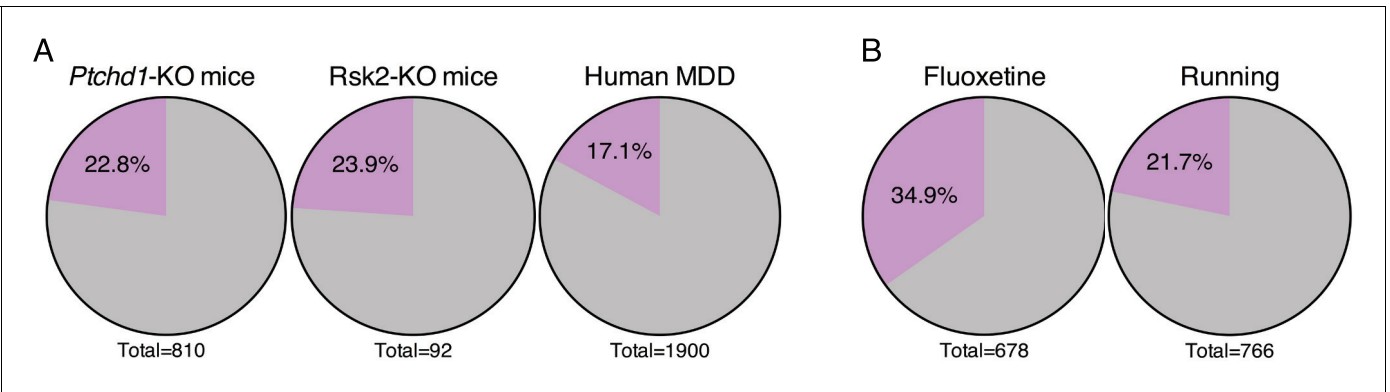

**Figure 7.** Forniceal DBS enhances the expression of genes relevant to other neuropsychiatric conditions. (**A**) Pie charts showing the overlap of DBS upregulated genes with genes found to be decreased in expression in the hippocampus of individuals with or mouse models of neuropsychiatric disorders. Gray: genes that are low in the indicated sample that are unchanged by DBS in WT mice (FDR > 0.05). Purple: genes with increased expression following DBS in WT mice (FDR < 0.05,>25% fold-change). The significance of the overlap between DBS and these datasets was determined using the Fisher exact test, yielding the following values: *Ptchd1*-KO vs. DBS: p=3.10E-25; Rsk2-KO vs. DBS: p=7.5E-6; MDD vs. DBS: p=4.3E-20. (**B**) Comparison of genes upregulated by DBS and genes upregulated by either fluoxetine or running. The significance of the overlap between DBS and these datasets was determined using the Fisher exact test, yielding the following values: fluoxetine vs. DBS: p=1.45E-67; running vs. DBS: p=3.7E-20. Comparison of genes upregulated by DBS and genes upregulated by either fluoxetine or running. Purple: genes that are increased by the indicated treatment and by DBS in WT mice (FDR < 0.05,>25% fold-change). Gray: genes increased by fluoxetine or running that are not increased by DBS in WT mice. *Figure 7—figure supplement 1* compares the genes that overlap with DBS (indicated by the purple slice in the pie chart) from each of the disorders shown in *Figure 7A*.

DOI: https://doi.org/10.7554/eLife.34031.024

The following figure supplement is available for figure 7:

**Figure supplement 1.** Overlap between the DBS dataset and neurologic disorders.

DOI: https://doi.org/10.7554/eLife.34031.025

p<0.001). These findings suggest that DBS utilizes some of the same transcriptional programs activated by fluoxetine and running.

## Discussion

The intricate dynamics of the neuronal response to stimulation are a pressing topic of research in neuroscience, while the use of neuromodulation is emerging as a viable intervention for many neuropsychiatric disorders. This work provides an unbiased snapshot of the molecular milieu of a neuron following DBS-induced activation and is the first study, to our knowledge, to combine proteomics, transcriptomics, and epigenomics to address the question of how neuronal activity, induced by an intervention used in humans, shapes the function of a neuron.

### The influence of forniceal DBS on wild-type neurons

We found that forniceal DBS induces changes in the expression of thousands of genes and splice variants, and many of these changes are in genes that have been previously reported to be activity-dependent (*Eom et al., 2013*; *Flavell et al., 2008*; *Halder et al., 2016*; *Kim et al., 2010*; *Lin et al., 2008*; *Madabhushi et al., 2015*; *Xiang et al., 2007*). The finding that many genes in the DBS dataset overlap with genes upregulated by environmental enrichment (see *Figure 1—figure supplement 1*) suggests that DBS acts through physiological pathways involved in plasticity.

The observation that a significant proportion of DBS-induced genes are involved in regulation of apoptosis is particularly interesting, as it hints that DBS is supporting the survival of either existing DG neurons or of newly born neurons in the subgranular zone. Consistent with the hypothesis that DBS influences newborn neurons is the finding that numerous genes are differentially expressed following DBS specifically in neuroprogenitor cells. This indicates that forniceal DBS not only influences gene expression in mature granule cells but in cycling cells in the dentate gyrus. This finding also helps explain why neurogenesis is strongly enhanced following chronic forniceal DBS (*Hao et al., 2015*). To better understand the mechanisms of action of forniceal DBS, further studies are needed to determine if neurogenesis is required for the behavioral improvement observed post-DBS. Such data may refine the patient populations (those with or without intact adult neurogenesis) that could benefit from this intervention.

Forniceal DBS also dramatically alters isoform expression in stimulated cells, and we identified hundreds of splicing changes, particularly in synaptic and neurogenic proteins, which occurred in the absence of overall gene-level expression changes. These changes are of particular note because they would be overlooked by analyses that focus on gene-level changes. Although many isoforms have yet to be characterized in terms of their function, localization, and/or interactor differences, many of the isoforms we detected do have unique features within their protein coding regions, suggesting that these isoforms play distinct roles from other isoforms of the same gene. More work is needed to better understand the functional consequences of these splicing changes and their overall influence on neurons.

Our proteomics data revealed that a subset of transcription factors are among the first proteins altered following DBS, including a number of Jun-family transcription factors. However, given the limitations in terms of the number of proteins that can be confidently identified in a given sample, we cannot rule out the possibility that some proteins may have been overlooked in our analysis. When looking at the targets of the identified transcription factors in the DBS dataset, we do find a number of genes and downstream transcription factors that have been directly linked to regulation by the Jun-family factors. This list of targets is still likely to underestimate the actual number of genes regulated by these transcription factors, as the TRANSFAC database, which contains validated interactions, is not an exhaustive list of all targets. These newly synthesized proteins help to augment the transcriptional cascade mediated by already existing Jun family proteins within the cell and may also serve as activity-dependent epigenetic regulators (*Su et al., 2017*). A recent study found that neuronal activity can influence chromatin accessibility and Junb and Fos are enriched at these newly opened regions (*Su et al., 2017*). Thus, these newly synthesized proteins may also act to reshape the chromatin landscape to influence later phases of activity-dependent gene expression.

In addition to transcription and proteomic changes, we also identified unique DNA methylation signatures of activity-dependent genes and thousands of DNA methylation changes that occur upon neuronal stimulation. Because the dentate gyrus is comprised primarily of granule neurons

(*Guo et al., 2011*), the methylation patterns we observed belong predominantly to this neuronal subtype. However, we cannot exclude the possibility that a subset of the DNA methylation changes we identified result from changes in other dentate cell types. We found that mCG levels tended to be lower on the promoters and gene bodies of DBS-upregulated genes. Further, mCG levels were lower on these genes in both sham and DBS samples, suggesting that lowered methylation provides a priming effect, enabling rapid gene expression increases. In addition to these preexisting methylation patterns that correlate with the response of the gene to DBS induction, we also noted thousands of regions where DBS induced methylation changes. Alterations in mCG density were overrepresented in exon and promoter regions, yet these changes did not tend to fall within the genes with the greatest alterations in expression post-DBS. An interesting possibility is that these methylation changes occur in genes whose expression rises at later time points. Unlike the changes in mCG methylation, mCH methylation changes tended to follow the genome distribution and lacked obvious methylation signatures on activity-dependent genes. These findings suggest that mCH may have less of a direct role in activity-dependent gene expression changes. It may be, however, that the changes we see in both mCG and mCH levels in intergenic regions affect the activity of enhancer regions, thereby indirectly influencing gene expression.

## Understanding the transcriptional effects of DBS in *Mecp2*-deficient mice

It is remarkable that nearly all of the same genes upregulated by DBS in WT mice were also upregulated in *Mecp2* KO mice—and to the same levels as those seen in WT animals. A point of open debate in the field is whether MeCP2 is involved in activity-dependent transcription (*Chen et al., 2003*; *Cohen et al., 2011*; *Ebert et al., 2013*; *Martinowich et al., 2003*; *Zhou et al., 2006*). These data unequivocally show that MeCP2 is not required for DBS-induced activity-dependent gene expression in the dentate gyrus: KO mice are capable of activating the same transcriptional programs following DBS as WT mice. This enhanced expression even occurs in genes such as *Bdnf*, which have historically been proposed to need changes in MeCP2 binding for appropriate expression after neuronal activation. The finding that MeCP2 is not required for DBS-induced transcription is also consistent with the methylation data we gathered. Genes that are upregulated by DBS have lower baseline mCG methylation levels. Thus, MeCP2 is less likely to bind to these genes and contribute to the regulation of their expression.

In addition to upregulating normal activity-dependent gene expression paradigms in KO mice, one of the likely benefits of DBS in *Mecp2*-null mice is the rescue of abnormal expression of synaptic proteins. Among the genes rescued are *Gad2* (Glutamate Decarboxylase 2) and *Grin2d* (Glutamate Ionotropic Receptor NMDA Type Subunit 2D). These genes are of particular interest because perturbations in the balance between inhibitory and excitatory signaling in MeCP2 mouse models has been well established, and both GABA synthesis and NMDA receptor signaling have been implicated in the circuit dysfunction observed in these mice (*Calfa et al., 2015*; *Chao et al., 2010*; *Chao et al., 2007*; *Ure et al., 2016*). This enhanced transcription and neurogenesis likely contribute to the improvements in learning and memory observed following chronic DBS in Rett mice. Consistent with this hypothesis are our findings of sustained transcriptional rescue in *Mecp2*-heterozygous female mice weeks after chronic forniceal DBS treatment ended. It is notable that 17 days after DBS was completed, we observed normalization of *Gad2* and a handful of other signaling molecules. While many genes show a trend toward rescued expression in Het DBS samples, we predict that more genes would be significantly rescued if we assessed transcription at an earlier time point post-chronic DBS, as the restoration of normal gene expression likely tapers off over time. Thus, repeating forniceal stimulation every few weeks throughout life would be necessary to maintain the effects of its benefits in dentate neurons.

## Potential beneficial effects of DBS in other neuropsychiatric disorders

Forniceal DBS holds therapeutic promise for other diseases where hippocampal function is disrupted, as we found that DBS-upregulated genes overlap significantly with genes downregulated in other intellectual disability disorders and depression. These findings suggest that SSRI-resistant patients with major depressive disorder may benefit from DBS, as this treatment is able to both

induce expression of genes that are downregulated in these patients as well as upregulate many of the same genes induced by fluoxetine treatment.

Finding effective treatments for intellectual disabilities is one of the most pressing unmet needs in modern medicine. The vast number of different genetic changes involved in intellectual disabilities and neuropsychiatric disorders makes it hard to imagine that a drug-based treatment could be effective. Although considered an invasive treatment, DBS has proven safe in humans and could offer therapeutic options for these otherwise untreatable disorders.

# Materials and methods

## Key resources table

| Reagent or resource | Source | Identifier |
|---|---|---|
| **Critical Commercial Assays** | | |
| Aurum Total RNA Fatty and Fibrous Tissue Kit | Bio-Rad | Cat #: 7326830 |
| TruSeq RNA Library Prep Kit v2 | Illumina | RS-122–2001 |
| Quick-DNA Universal Kit | ZymoResearch | Catalog #: D4068 |
| EZ DNA Methylation Lightning Kit | ZymoResearch | Catalog #: D5030 |
| TruSeq DNA Methylation Kit | Illumina | EGMK81312 |
| **Deposited Data** | | |
| Raw data | This paper | GSE107357, GSE107383 and GSE111703 |
| Analyzed data | This paper | See Source Data files |
| Mouse reference genome GRCm38 (M10) | GENCODE | http://www.gencodegenes.org/mouse_releases/10.html |
| Duric (MDD dataset) | *Duric et al. (2013)* | GSE24095 |
| Madabhushi (Activity Dependent Genes) | *Madabhushi et al. (2015)* | GSE61887 |
| Reanalyzed Running Dataset (Neurogenesis) | *Miller et al. (2013)* | GSE39697 |
| Flavell (Activity Dependent Genes) | *Flavell et al. (2008)* | GSE13539 |
| Guo (Activity Dependent Genes) | *Guo et al. (2011)* | GSE30493 |
| Halder (Activity Dependent Genes) | *Halder et al. (2016)* | GSE74971 |
| Kim (Activity Dependent Genes) | *Kim et al. (2010)* | GSE21161 |
| Lacar (Activity Dependent Genes) | *Lacar et al. (2016)* | GSE77067 |
| Lin (Activity Dependent Genes) | *Lin et al. (2008)* | GSE11261 |
| Xiang (Activity Dependent Genes) | *Xiang et al. (2007)* | GSE6254 |
| Fluoxetine (Neurogenesis) | *Samuels et al. (2014)* | GSE43261 |
| Rsk2 KO/Coffin Lowry Syndrome (IDD) | *Mehmood et al. (2011)* | GSE22137 |
| PTCHD1 KO (IDD) | *Ung et al. (2017)* | GSE80312 |
| **Experimental Models: Organisms/Strains** | | |
| Mecp2-null (KO) mouse (B6.129P2(C)-Mecp2<sup>tm1.1Bird</sup>/J) | Jackson laboratory | Stock number: 003890 RRID:IMSR_JAX:003890 |
| Wild-type FVB males for breeding (FVB/NJ) | Jackson laboratory | Stock number: 001800 RRID:IMSR_JAX:001800 |
| Wild-type 129 males for backcrossing (129S6/SvEvTac) | Taconic | Catalog #: 129SVE-M RRID:IMSR_TAC:129sve |
| **Oligonucleotides** | | |
| Primers for RT-qPCR | This paper | *Supplementary file 1* |

*Continued on next page*

*Continued*

| Reagent or resource | Source | Identifier |
|---|---|---|
| **Recombinant DNA** | | |
| Unmethylated cl857 Sam7 Lambda DNA | Promega | D1521 |
| **Software and Algorithms** | | |
| STAR aligner (v2.4.2a) | *Dobin et al. (2013)* | https://github.com/alexdobin/STAR/releases/tag/STAR_2.4.2a RRID:SCR_015899 |
| HTSeq (v 0.9.1) | (*Anders et al., 2015*) | https://htseq.readthedocs.io/en/release_0.9.1/ RRID:SCR_005514 |
| DESeq2 | *Love et al. (2014)* | https://bioconductor.org/packages/release/bioc/html/DESeq2.html RRID:SCR_015687 |
| ggplot2 | Hadley Wickham. ggplot2: Elegant Graphics for Data Analysis (2010) | https://github.com/tidyverse/ggplot2 RRID:SCR_014601 |
| UpSetR | CRAN Package | https://cran.r-project.org/web/packages/UpSetR/index.html https://doi.org/10.1093/bioinformatics/btx364 RRID:SCR_003005 |
| Limma | Bioconductor Package | http://bioconductor.org/packages/release/bioc/html/limma.html RRID:SCR_010943 |
| pheatmap | CRAN Package | https://cran.r-project.org/web/packages/pheatmap/index.html RRID:SCR_003005 |
| GeneOverlap | (*Shen, 2013*) | http://bioconductor.org/packages/GeneOverlap/ RRID:SCR_006442 |
| TRANSFAC database | geneXplain | http://genexplain.com/transfac/ RRID:SCR_005620 |
| Kallisto | (*Bray et al., 2016*) | https://pachterlab.github.io/kallisto/download |
| Sleuth | (*Pimentel et al., 2017*) | https://github.com/pachterlab/sleuth |
| R Project for Statistical Computing | https://www.r-project.org | RRID:SCR_001905 |
| Population Specific Expression Analysis | (*Kuhn et al., 2011*) | https://bioconductor.org/packages/release/bioc/html/PSEA.html RRID:SCR_006442 |
| rMATS v3.2.5 | (*Shen et al., 2014*) | http://rnaseq-mats.sourceforge.net/ |
| Bismark | (*Krueger and Andrews, 2011*) | http://www.bioinformatics.babraham.ac.uk/projects/bismark/ RRID:SCR_005604 |
| Prism 6 | https://www.graphpad.com/scientific-software/prism/ | RRID:SCR_015807 |
| Tableau Desktop 10.5 | https://www.tableau.com | RRID:SCR_013994 |
| methylKit | (*Akalin et al., 2012*) | https://bioconductor.org/packages/release/bioc/html/methylKit.html RRID:SCR_005177 |
| deepTools | (*Ramírez et al., 2014*) | http://deeptools.readthedocs.io/en/latest/index.html |
| IGV | (*Thorvaldsdóttir et al., 2013*) | http://software.broadinstitute.org/software/igv/ RRID:SCR_011793 |
| webGestalt | (*Zhang et al., 2005*) | http://www.webgestalt.org/option.php RRID:SCR_006786 |

*Continued on next page*

*Continued*

| Reagent or resource | Source | Identifier |
|---|---|---|
| EntichmentMap | (*Merico et al., 2010*) | http://baderlab.org/Software /EnrichmentMap |
| Cytoscape | (*Shannon et al., 2003*) | http://cytoscape.org/ RRID:SCR_003032 |
| bedGraphToBigWig | ENCODE | https://www.encodeproject.org/ software/bedgraphtobigwig/ RRID:SCR_015482 |

## Contact for reagent and resource sharing

Further information and requests for resources and reagents should be directed to and will be fulfilled by the Lead Contact, Huda Zoghbi (hzoghbi@bcm.edu).

## Experimental model and subject details

### Mice

The Mecp2-null mice were obtained from Jackson laboratory (Strain: B6.129P2(C)-Mecp2$^{tm1.1Bird/J}$; stock number: 003890). Heterozygous females were back-crossed to pure 129 males for >10 generations. These pure 129 *Mecp2*-heterozygous females were then bred with wild-type FVB males to produce the FVB/129 F1-hybrid mice that were used for all of the studies in this paper. They were group-housed with up to five mice per cage prior to surgery and individually housed with nesting material in the cage after surgery. They were maintained on a 14 hr light:10 hr dark cycle (light on at 06:00) with standard mouse chow and water *ad libitum* in our AAALAS-accredited facility. All experimental procedures and tests were conducted during the light cycle. Surgery was performed on 6–7 week-old male wild-type and *Mecp2$^{-/Y}$* mice, and 6–7 week-old female wild-type and *Mecp2*-heterozygous mice. The evoked activity in the dentate gyrus was reevaluated two weeks after electrode placement, and the strength of the elicited activity was used to assign mice to either the DBS (strong signal) or sham groups (weak signal). Acute DBS or sham treatment was performed between two and three weeks post-surgery (mice were 8–9 weeks of age). Chronic DBS was started when mice were approximately 9 weeks of age.

### Study approval

All research and animal care procedures were approved by the Baylor College of Medicine Institutional Animal Care and Use Committee.

## Method details

### Deep Brain Stimulation

Surgery was performed as previously described (*Hao et al., 2015*). Mice were unilaterally implanted at 6 weeks of age with a bipolar stimulating electrode directed to the fimbria-fornix (FF) and two recording electrodes placed in the CA1 and dentate gyrus regions of the hippocampus, respectively. After a two-week recovery period, the mice were assigned to either the DBS or the sham group, and acute DBS or sham treatment was performed in 8–9 weeks old mice. Both groups were exposed to identical conditions except the sham group did not receive electrical stimulation. Acute DBS parameters: 45 min of DBS (130 Hz, 60µs, 50µA) with a 20 min recovery period prior to euthanasia. DG tissue was microdissected in ice cold PBS, flash frozen and stored at −80°C until it was needed for downstream studies. For chronic DBS, mice were also implanted at 6 weeks of age, given two weeks to recover, and then assigned to either DBS or sham treatment. Chronic DBS parameters: 1 hr per day for 14 days (130 Hz, 60µs, 50µA). Sham mice were concurrently placed into stimulation chambers for 1 hr daily but did not receive electrical stimulation. Dentate gyrus tissue was collected 17 days after end of the 2 weeks of stimulation or sham treatment.

### RNA isolation

RNA extraction and purification was performed using the Aurum Total RNA Fatty and Fibrous Tissue Kit (Bio-Rad, Hercules, CA; catalog #: 7326830) per the kit instructions, and genomic DNA was

eliminated using an on-column DNase digestion step. RNA quality was assessed using the Agilent 2100 Bioanalyzer system prior to being used for either library preparation (deep sequencing) or reverse transcription (qPCR).

## RNA-Sequencing

RNA library preparation for the acute DBS male samples was performed using three sham-treated and 4 DBS-treated *Mecp2*-null mice (biological replicates), and two sham-treated and 4 DBS-treated wild-type mice (biological replicates) were used for library preparation, and one wild-type sham sample was sequenced twice to provide a third technical replicate so that counts for sham gene expression would be similar between the two genotypes. For the chronic DBS female samples, 4 WT sham and 4 WT DBS mice were used (biological replicates) and 3 *Mecp2*-heterozygous sham mice and four heterozygous DBS mice were used for sequencing (biological replicates). For both acute DBS and chronic DBS samples, the TruSeq RNA Library Prep Kit v2 (Illumina RS-122–2001) and 250 ng of total RNA was used to make each library, and ERCC RNA Spike-In Mix 1 or 2 (Thermo Fisher 4456740) was added to the samples prior to library preparation. Samples were paired end sequenced (PE100) on a HiSeq2500 in high-output mode using the v4 sequencing kit (Illumina; SBS: FC-401–4003, paired-end flow cell: PE-401–4001).

## Reverse transcription and qPCR

For the acute DBS male data, a new cohort of mice was used for the qPCR analysis, with four sham-treated and 4 DBS-treated mice per genotype (each sample is a biological replicate). Additionally, for the acute female DBS data, a new cohort was used that had 4 WT sham, 4 WT DBS, 4 Het sham, and 6 Het DBS samples (each a biological replicate; age 8–9 weeks old at time of tissue collection). Extracted total RNA was reverse transcribed using M-MLV Reverse Transcriptase (Invitrogen, Waltham, MA; catalog #: 28025013) and random primers (Invitrogen 48190011). Quantitative PCR was performed using iTaq Universal SYBR Green Supermix (Bio-Rad 1725124), and a CFX96 Real-Time PCR Detection System (Bio-Rad C1000 Thermal Cycler). Cycling conditions: 1. 95°C for 3 min 2. 95°C for 0:10 3. 58°C for 0:30 + plate read 4. Go to 2, 39 more times 5. 95°C for 0:10 6. Melt curve 65°C to 95°C, increment of 0.5°C for 0:05 + plate read. The primers used for PCR are listed in *Supplementary file 1*.

## Bisulfite library preparation and sequencing

Genomic DNA was extracted from two wild-type male sham mice (biological replicates) and two wild-type male DBS mice (biological replicates) using the Quick-DNA™ Universal Kit (ZymoResearch catalog #: D4068) according to the kit instructions for solid tissues. 50 ng of extracted gDNA were spiked with 0.1% (w/w) of Unmethylated cl857 Sam7 Lambda DNA (Promega, Madison, WI) that served as an unmethylated internal control. The DNA mixture was subjected to bisulfite treatment (ZymoResearch EZ DNA Methylation Lightning Kit; ZymoResearch, Irvine, CA) and was eluted into 9 uL. Library generation from bisulfite treated DNA was performed using the Illumina TruSeq DNA Methylation Kit (Illumina, San Diego, CA) according to the manufacturer's instructions. To allow for multiplexed sequencing of libraries from bisulfite-treated DNA, individual libraries were barcoded using the Illumina TruSeq DNA Methylation Index PCR kit (Illumina, San Diego, CA; Catalog #: EGIDX81312). Samples were then sent to the Genomic and RNA Profiling Core (GARP) for sequencing.

The Genomic and RNA Profiling (GARP) Core first conducted Sample Quality checks using the NanoDrop spectrophotometer and Agilent Bioanalyzer 2100. To quantify the adapter-ligated library and confirm successful P5 and P7 adapter incorporations, we used the Applied Biosystems ViiA7 Real-Time PCR System and a KAPA Illumina/Universal Library Quantification Kit. Additional bead clean-up using Beckman Coulter Agencourt AMPure XP beads was needed to remove residual primer/adapter dimers prior to quantification and sequencing. GARP then sequenced the libraries on the HiSeq 2500 Sequencing System using the High Output v4 System.

Briefly, to remove residual primer dimers, measured at 6% by pmol/ul on the Agilent 2100 BioAnalyzer, GARP performed an additional post-library prep clean up using Beckman Coulter Agencourt AMPure XP Beads (p/n A63881) at a 1:1 library to bead volume proportion. The manufacturer's protocol was then used to remove the lower molecular weight fragments which include previous

unremoved primer dimers. The resulting libraries were quantitated using the NanoDrop spectrophotometer and fragment size assessed with the Agilent Bioanalyzer to confirm dimer removal. A qPCR quantitation was performed on the libraries to determine the concentration of adapter ligated fragments using Applied Biosystems ViiA7 Real-Time PCR System and a KAPA Illumina/Universal Library Quantification Kit (product p/n; KK4824, protocol v1.14). Libraries were then equimolarly pooled and the resulting pool was again quantified using the KAPA Library Quantification Kit. Using the concentration from the ViiA7 qPCR machine above, 27pM and 28pM of library was loaded onto thirteen lanes on two high output v4 flowcells (Illumina p/n PE-401–4001) and amplified by bridge amplification using the Illumina cBot machine (cBot protocol: PE_HiSeq_Cluster_Kit_v4_cBot_recipe_v9.0). PhiX Control v3 adapter-ligated library (Illumina p/n 15017666) and RNA-Seq library was spiked-in at 2% and 13% by weight, respectively, to ensure sufficient balanced diversity and to monitor clustering and sequencing performance. A paired-end 125 cycle run was used to sequence the flowcell on a HiSeq 2500 Sequencing System (Illumina p/n FC-401–4003).

## Sample preparation for mass spectrometry

Wild-type male samples were used for proteomics analyses. There were 5 DBS samples (biological replicates) and four sham samples (biological replicates). Samples were prepared as previously described (*Weekes et al., 2014*) with the following modification. All solutions are reported as final concentrations. Lysis buffer (8 M Urea, 1% SDS, 50 mM Tris pH 8.5, Protease and Phosphatase inhibitors from Roche) was added to the cell pellets to achieve a cell lysate with a protein concentration of ~2 mg/mL and the final concentration was determined by a micro-BCA assay (Pierce, Rockford, IL). Proteins were reduced with 5 mM DTT at room temperature for one hour and alkylated with 15 mM Iodoacetamide at room temperature for one hour in the dark. Proteins were precipitated using methanol/chloroform. In brief, four volumes of methanol was added to the cell lysate, followed by one volume of chloroform, and three volumes of water. The mixture was vortexed and centrifuged to separate the chloroform phase from the aqueous phase. The precipitated protein was washed with one volume of ice cold methanol. The washed precipitated protein was allowed to air dry. Precipitated protein was resuspended in 4 M Urea, 50 mM Tris pH 8.5. Proteins were first digested with LysC (1:50; enzyme:protein) for overnight at 25°C. The LysC digestion was diluted to 1 M Urea, 50 mM Tris pH 8.5 and then digested with trypsin (1:100; enzyme:protein) for another 8 hr at 25°C. Peptides were desalted using a $C_{18}$ solid phase extraction cartridges as previously described. Dried peptides were resuspended in 200 mM EPPS, pH 8.0. Peptide quantification was performed using the micro-BCA assay (Pierce). The same amount of peptide from each condition was labeled with tandem mass tag (TMT) reagents (1:4; peptide:TMT label) (Pierce). The 10-plex labeling reactions were performed for 2 hr at 25°C. Modification of tyrosine residues with TMT was reversed by the addition of 5% hydroxyl amine for 15 min at 25°C. The reaction was quenched with 0.5% TFA and samples were combined at a 1:1:1:1:1:1:1:1:1:1 ratio. Combined samples were desalted and offline fractionated into 24 fractions as previously described.

## Liquid chromatography-MS3 spectrometry (LC-MS/MS)

12 of the 24 peptide fraction from the basic reverse phase step (every other fraction) were analyzed with an LC-MS3 data collection strategy (*McAlister et al., 2014*) on an Orbitrap Fusion mass spectrometer (Thermo Fisher Scientific, Waltham, MA) equipped with a Proxeon Easy nLC 1000 for online sample handling and peptide separations. Approximately 5 µg of peptide resuspended in 5% formic acid +5% acetonitrile was loaded onto a 100 µm inner diameter fused-silica micro capillary with a needle tip pulled to an internal diameter less than 5 µm. The column was packed in-house to a length of 35 cm with a $C_{18}$ reverse phase resin (GP118 resin 1.8 µm, 120 Å, Sepax Technologies, Newark, DE). The peptides were separated using a 180 min linear gradient from 3% to 25% buffer B (100% ACN +0.125% formic acid) equilibrated with buffer A (3% ACN +0.125% formic acid) at a flow rate of 600 nL/min across the column. The scan sequence for the Fusion Orbitrap began with an MS1 spectrum (Orbitrap analysis, resolution 120,000, 400–1400 m/z scan range, AGC target $2 \times 10^5$, maximum injection time 100 ms, dynamic exclusion of 90 s). The 'Top10' precursors were selected for MS2 analysis, which consisted of CID (quadrupole isolation set at 0.5 Da and ion trap analysis, AGC $8 \times 10^3$, NCE 35, maximum injection time 150 ms). The top ten precursors from each MS2 scan were selected for MS3 analysis (synchronous precursor selection), in which precursors

were fragmented by HCD prior to Orbitrap analysis (NCE 55, max AGC $1 \times 10^5$, maximum injection time 150 ms, isolation window 2.5 Da, resolution 60,000.

## Quantification and statistical analysis

### Sample size selection

Data collection and experimental analyses were performed in a blinded manner whenever possible. Sample sizes for RNA and protein analyses were chosen based on prior studies from our laboratory to ensure adequate power for statistical analyses. Sample number and coverage for whole-genome bisulfite sequencing was determined based on the recommendations in *Ziller et al., 2015* (*Ziller et al., 2015*).

### Differential gene expression analyses

The quality of raw reads was assessed using FastQC (*Andrews, 2010*). The average per base quality score across all the files were greater than 34 and had passed all the major tests. The raw reads were aligned to the Mus musculus genome (Gencode Version M10 – Ensembl 85) using STAR v2.4.2a (*Dobin et al., 2013*). The mappability of unique reads for each sample was approximately 92%. The raw counts were computed using *quantMode* function in STAR and HTSeq (*Anders et al., 2015*) for chronic DBS data. The obtained read counts are analogous to the expression level of each gene across all the samples.

RNA libraries for the acute DBS male samples were prepared in two batches, and sequencing was performed in three batches. Therefore, we used batch as a covariate in our model for differential expression analysis using DESeq2 (*Love et al., 2014*). RNA libraries from the chronic DBS female samples were prepared in a single batch and were sequenced in a single batch. Genes with raw mean reads greater than 40 (i.e., ~15,568 genes) were used for normalization and differential gene expression analysis using DESeq2 package in R. Wald test defined in the DESeq function of the package was used for differential expression analysis and shrunken log fold-changes (i.e., obtaining reliable variance estimates by pooling information across all the genes) were used for further analysis. For chronic DBS data all protein coding genes with mean read count greater than 40 were tested for differential expression. Principal Component Analysis (PCA), hierarchical clustering plots (hclust) and XY plots between replicates of same genotypes were used to examine for nominal amounts of non-technical variation and other latent factors. PCA, hclust and bar plots were generated using ggplot2 (*Wickham, 2010*). All the heatmaps of differentially expressed genes were generated using pheatmap package (*Kolde, 2012*) in R environment, except for heat maps shown in *Figure 6C* which were generated using Tableau Desktop 10.5.

### Comparison of RNA-Seq data to other activity datasets

To determine the overlap between upregulated genes due to DBS at baseline and with the genes related to activity, intellectual disorder and neurogenesis, the published list of differentially expressed genes were downloaded from the supplementary files in each study except for GSE39697 (or Running Dataset on GEO [*Barrett et al., 2013*]). Because of the frequent changes in gene name and annotation, we used MGI batch query (*Eppig et al., 2015*) to facilitate uniform comparison between these gene lists. Statistical significance of association for each of the overlapping gene lists was calculated using *testGeneOverlap* function in GeneOverlap package (*Shen, 2013*). *testGeneOverlap* function uses Fisher's exact test to find the significance.

For comparison of activity dependent genes, we used the list of differentially expressed upregulated genes (*Flavell et al., 2008*; *Guo et al., 2011*; *Halder et al., 2016*; *Kim et al., 2010*; *Lacar et al., 2016*; *Lin et al., 2008*; *Xiang et al., 2007*) and downregulated genes (*Madabhushi et al., 2015*). We used the list of differentially expressed upregulated genes from following studies for comparison of neurogenesis genes (*Lopez-Atalaya et al., 2011*; *Samuels et al., 2014*). In case of running dataset (GSE39697), we compared mice samples across two different conditions – 4 days of running and zero running. RMA function (*Gautier et al., 2004*; *Irizarry et al., 2003*) in the R 'affy' package was used to perform background correction, normalization and summarization of core probesets. NetAffx annotation file was used to map affy probes to its official gene symbols. Limma (*Ritchie et al., 2015*), a R package was used for differential expression analysis and genes with FDR < 0.05 and absolute logFC >0 were called as differentially expressed and were

used for further analysis. For Intellectual Disorder, we used list of differentially expressed downregulated genes from the following studies (*Mehmood et al., 2011*; *Ung et al., 2017*).

## Differentially Expressed Isoforms analyses

DEIs were computed using the pipeline described in (*Yalamanchili et al., 2017*). Isoform expression was quantified from raw pair-end fastq files (RNA-Seq data) using *Kallisto* (*Bray et al., 2016*), an alignment free transcript quantification program. Reference transcript sequences of GRCm38 VM10 were downloaded from GENCODE website. Kallisto index was build using a kmer length of 31 and quantification was performed using 100 bootstrap samples. Differentially expressed Isoforms were computed using both *DESeq2* (*Love et al., 2014*) and *Sleuth* (*Pimentel et al., 2017*) independently. Batch information is passed as a covariate. An FDR cutoff of 0.05 and fold change cutoff of 2 were used to call significant DEIs. For reliability, only isoforms that were called differential by both the methods (DESeq2 and Sleuth) were considered for further downstream analyses.

## RNA splicing analysis

Annotated alternative splicing changes were quantified and classified using rMATS (*Shen et al., 2014*). Alignment files (BAM) and reference annotations (GTF) from GENCODE were passed to rMATS. Insert length is computed as average fragment size (400 bp) - (2*read length). rMATS classifies splicing events into five categories, skipped exons, retained introns, mutually exclusive exons, alternative 5′ and 3′ splice sites. An FDR cutoff of 0.05 and an inclusion level difference cutoff of less than −0.2 or greater than 0.2 were used to screen for statistically significant changes.

## DNA methylation analyses

Reference genome preparation and read alignment were done using Bismark (*Krueger and Andrews, 2011*). Bisulfite treatment compatible reference (mm10) was prepared by converting C→T and G→A using bismark_genome_preparation command. Raw fastq files were aligned to the bisulfite-converted reference using the command bismark using bowtie2 (*Langmead and Salzberg, 2012*) aligner with multi-seed length of 25 and 0 mismatch tolerance (-L 25 N 0). Next, methylated C's were called in three different contexts (CpG, CHG and CHH) using bismark_methylation_extractor command. The output from methylation extractor was then converted to bedgraph format using bismark2bedGraph script. Bedgraph files were visualized using Integrative Genomics Viewer (*Thorvaldsdóttir et al., 2013*) Sample-wise genome-wide cytosine reports with every single cytosine in the genome were generated using the coverage2cytosine module. Cytosine positions with a minimum of 10 mapped reads were considered for further downstream analysis.

Base level differential methylation was computed using methylKit (*Akalin et al., 2012*). Samples were normalized to median coverage and replicates were pools to perform fisher's test. Significant differentially methylated regions (DMRs) were extracted with an q-value cutoff of 0.05% and 50% methylation difference. These DMRs were annotated gene features such as promoter, exon, intron and intergenic accordingly. Significance levels were estimated by random sampling genomic regions (as many as DMRs) 1000 times and annotating them to respective regions. P-values were computed as the proportion of random trials that were greater/less than the observed proportions.

Methylation profiles (mCG and mCH) of up-regulated, down-regulated and unaltered genes (from RNA-Seq data) were plotted using deeptools (*Ramírez et al., 2014*). ComputeMatrix module was used to quantify the methylation (from bedgraph files) and plotProfile module was used to draw the line plots. Methylation levels for differentially spliced exons were computed at 5 p and 3 p ends as a window of 300 bp (250 intronic and 50 exonic nucleotides). Base level methylation was computed as the proportion of number of methylated cytosines to the total number of reads (cytosines) aligned to that position. Methylation level of a region is computed as the average base level methylation of constituting cytosine positions.

## Gene Ontology analysis

Gene ontology term enrichment analysis was performed using WEB-based Gene SeT AnaLysis Toolket (*Zhang et al., 2005*). All expressed genes with mean read count greater than 40 were used as background (a total of 13785 genes). An FDR cutoff of 0.05 was used. Top GO terms were clustered

based on the number of genes shared using EnrichmentMap (*Merico et al., 2010*) and visualized in Cytoscape (*Shannon et al., 2003*). Bar plots were generated using the Prism six software package.

## Network analysis

To study the regulatory networks altered due to DBS, we utilized the TRANScription FACtor database (TRANSFAC, release 2017.2), specific to interactions annotated to mouse. First, we extracted the target genes of the three transcriptional factors (TFs) that were found to be increased in the mass spectrometry data (Jun, Junb, and Fosl2). Next, we further extracted the targets of genes extracted from the first step. This two-step process reveals both the direct (first layer) and strong indirect (the second layer) transcriptional changes of these three TFs. Finally, we overlaid the expression change patterns observed at the gene- and isoform-levels on the extracted regulatory network. In *Figure 3*, we showed the core regulatory network, which is composed of genes that are either significant DEG's (FDR < 0.05, 2-fold) or DEI's (FDR < 0.05, 2-fold).

## PSEA analysis

PSEA (Population Specific Expression Analysis) (*Kuhn et al., 2011*) was used to estimate cell type specific gene expression changes. Four regression models have been built for the major cellular subpopulations inside the dentate gyrus using marker genes highly expressed in granular cells ('C1ql2','Dsp','Trpc6','Pitpnm2','Btg1'), mossy cells ('Calb2','Fgf1'), neural stem cells ('Nes','Sox2','Fabp7'), or neural progenitor cells ('Dcx','Dpysl3'). Marker genes used to calculate the reference expression signal were selected based on published cellular markers. Experimental conditions (DBS vs. Sham) were used to build the auxiliary regressor to retrieve cell type specific differential expression genes. The cell-type specific differentially expressed genes were selected based on PSEA optimal regression model with the following criteria: F-test, $p<0.01$, adjusted R squared >0.8. To reduce the number of false positive results, genes with large intercepts were excluded. All of the above deconvolution analyses were performed through the PSEA R package (PSEA version: 1.8.0; www.bioconductor.org/packages/release/bioc/html/PSEA.html). PSEA results are available in *Figure 1—source data 3*, which provides the following information: coef.1 is the noise, coef.2 is the normalized population specific expression, coef.3 is the normalized relative differences between DBS and sham treatment, and pvalue.1, pvalue.2, and pvalue.3 indicate significance of coef.1, 2, or 3, respectively, as determined by the F-test in the linear regression analysis.

## RT-qPCR analysis

Samples were run in triplicate for all primer sets analyzed, and the average $C_t$ value of the three replicate wells was used for the relative quantification of gene expression using the comparative $C_t$ method (*Schmittgen and Livak, 2008*). With this method, the average $C_t$ value for a given primer was compared to the geometric mean of the $C_t$ values of three control primers: mGAPDH, mActB, and mAtf2. These control primers were chosen based on the low variability in expression level between wild-type and *Mecp2*-null male mice as well as low variability between the male sham and DBS groups (*Figure 1—figure supplement 2*). Significance of expression was determined by using an unpaired, two-tailed t-test comparing the expression in DBS samples vs the expression in sham samples for a given genotype or with an two-way ANOVA with Holm-Sidak correction for multiple comparisons when comparing heterozygous sham, Het DBS, WT sham, and WT DBS mice using the GraphPad Prism six software; bar plots were also generated using the Prism six software. Primers for RT-qPCR can be found in *Supplemental file 1*.

## LC-MS3 data analysis

A suite of in-house software tools were used to for. RAW file processing and controlling peptide and protein level false discovery rates, assembling proteins from peptides, and protein quantification from peptides as previously described. MS/MS spectra were searched against a Uniprot mouse database (February 2014) with both the forward and reverse sequences. Database search criteria are as follows: tryptic with two missed cleavages, a precursor mass tolerance of 50 ppm, fragment ion mass tolerance of 1.0 Da, static alkylation of cysteine (57.02146 Da), static TMT labeling of lysine residues and N-termini of peptides (229.162932 Da), and variable oxidation of methionine (15.99491 Da). TMT reporter ion intensities were measured using a 0.003 Da window around the theoretical m/

z for each reporter ion in the MS3 scan. Peptide spectral matches with poor quality MS3 spectra were excluded from quantitation (<200 summed signal-to-noise across 10 channels and <0.5 precursor isolation specificity). Significance of protein level differences was calculated using an unpaired, two-tailed t-test.

## Data and software availability
### Sequencing data availability
The raw reads from the RNA-Sequencing and bisulfite sequencing data can be accessed on GEO with the following record numbers: GSE107357, GSE107383, and GSE111703. Our quantified gene and transcript counts following mapping can be accessed in *Figure 1—source data 1*, *Figure 2—source data 1*, *Figure 5—source data 1*, *Figure 5—source data 2*, and *Figure 6—source data 1*. All quantified mass spectrometry data is available in *Figure 3—source data 1*, and the locations of significant differentially methylated regions (DMRs) for each methylation type (mCG and mCH) are provided in *Figure 4—source data 1*.

### TRANSFAC database software
A license was purchased from geneXplain (http://genexplain.com/transfac/) for use of this database.

## Acknowledgements
This work supported by the Thermo Fisher Scientific Center for Multiplexed Proteomics at Harvard Medical School and by the Genomic and RNA Profiling Core at Baylor College of Medicine and the expert assistance of the core director, Dr. Lisa D White, Ph.D. This project was funded by the NIH (5R01NS057819 to HYZ), the Howard Hughes Medical Institute (HYZ), the Robert and Janice McNair Foundation (AEP), and Baylor Research Advocates for Student Scientists (AEP). This work was also made possible by the Baylor College of Medicine Intellectual and Developmental Disabilities Research Center Neuroconnectivity Core (NIH, U54 HD083092 from the National Institute of Child Health and Human Development). We thank members of the Zoghbi laboratory and Vicky Brandt for critical input on this manuscript and Zhenyu Wu and Sara Fares for their assistance on this project.

## Additional information

### Competing interests
Huda Y Zoghbi: Senior editor, *eLife*. The other authors declare that no competing interests exist.

### Funding

| Funder | Grant reference number | Author |
| --- | --- | --- |
| National Institutes of Health | 5R01NS057819 | Huda Y Zoghbi |
| Howard Hughes Medical Institute | HHMI Investigator | Huda Y Zoghbi |
| Robert and Janice McNair Foundation | Student Scholar | Amy E Pohodich |
| Baylor Research Advocates for Student Scientists | Student Scholar | Amy E Pohodich |

The funders had no role in study design, data collection and interpretation, or the decision to submit the work for publication.

### Author contributions
Amy E Pohodich, Conceptualization, Formal analysis, Funding acquisition, Validation, Investigation, Visualization, Methodology, Writing—original draft, Writing—review and editing; Hari Yalamanchili, Ayush T Raman, Software, Formal analysis, Validation, Investigation, Visualization, Writing—review and editing; Ying-Wooi Wan, Software, Formal analysis, Investigation, Visualization, Writing—review

and editing; Michael Gundry, Resources, Validation, Investigation, Writing—review and editing; Shuang Hao, Investigation, Writing—review and editing; Haijing Jin, Software, Formal analysis, Writing—review and editing; Jianrong Tang, Resources, Writing—review and editing; Zhandong Liu, Resources, Software, Supervision, Writing—review and editing; Huda Y Zoghbi, Conceptualization, Resources, Supervision, Funding acquisition, Methodology, Writing—review and editing

### Author ORCIDs
Amy E Pohodich http://orcid.org/0000-0002-1802-7995
Huda Y Zoghbi http://orcid.org/0000-0002-0700-3349

### Ethics
Animal experimentation: This study was performed in strict accordance with the recommendations in the Guide for the Care and Use of Laboratory Animals of the National Institutes of Health. All research and animal care procedures were approved by the Baylor College of Medicine Institutional Animal Care and Use Committee (approved protocols: AN-1013 and AN-5585). All surgery was performed under isoflurane anesthesia, and every effort was made to minimize pain and suffering.

### Decision letter and Author response
Decision letter https://doi.org/10.7554/eLife.34031.061
Author response https://doi.org/10.7554/eLife.34031.062

## Additional files

### Supplementary files
• Supplementary file 1. RT-qPCR primers used for validation of gene expression and splicing differences.
DOI: https://doi.org/10.7554/eLife.34031.026
• Transparent reporting form
DOI: https://doi.org/10.7554/eLife.34031.027

### Major datasets
The following datasets were generated:

| Author(s) | Year | Dataset title | Dataset URL | Database, license, and accessibility information |
|---|---|---|---|---|
| Pohodich AE, Zoghbi HY | 2018 | RNA-Sequencing data - acute DBS | https://www.ncbi.nlm.nih.gov/geo/query/acc.cgi?acc=GSE107357 | Publicly available at the NCBI Gene Expression Omnibus (accession no: GSE107357). |
| Pohodich AE, Zoghbi HY | 2018 | Whole-Genome bisulfite sequencing | https://www.ncbi.nlm.nih.gov/geo/query/acc.cgi?acc=GSE107383 | Publicly available at the NCBI Gene Expression Omnibus (accession no: GSE107383). |
| Pohodich AE, Zoghbi HY | 2018 | RNA-Sequencing data - chronic DBS | https://www.ncbi.nlm.nih.gov/geo/query/acc.cgi?acc=GSE111703 | Publicly available at the NCBI Gene Expression Omnibus (accession no: GSE111703). |

The following previously published datasets were used:

| Author(s) | Year | Dataset title | Dataset URL | Database, license, and accessibility information |
|---|---|---|---|---|
| Duric V, Banasr M, Stockmeier CA, Simen AA, Newton SS, Overholser JC, Jurjus GJ, Dieter L, Duman RS | 2013 | Altered expression of synapse and glutamate related genes in post-mortem hippocampus of depressed subjects. | https://www.ncbi.nlm.nih.gov/geo/query/acc.cgi?acc=GSE24095 | Publicly available at the NCBI Gene Expression Omnibus (accession no: GSE24095). |
| Miller JA, Nathanson J, Franjic D, Shim S, Dalley RA, Shapouri S, Smith KA, Sunkin SM, Bernard A, Bennett JL, Lee CK, Hawrylycz MJ, Jones AR, Amaral DG, Sestan N, Gage FH, Lein ES | 2013 | Conserved molecular signatures of neurogenesis in the hippocampal subgranular zone of rodents and primates | https://www.ncbi.nlm.nih.gov/geo/query/acc.cgi?acc=GSE39697 | Publicly available at the NCBI Gene Expression Omnibus (accession no: GSE39697). |
| Flavell SW, Kim TK, Gray JM, Harmin DA, Hemberg M, Hong EJ, Markenscoff-Papadimitriou E, Bear DM, Greenberg ME | 2008 | Genome-wide analysis of MEF2 transcriptional program reveals synaptic target genes and neuronal activity-dependent polyadenylation site selection | https://www.ncbi.nlm.nih.gov/geo/query/acc.cgi?acc=GSE13539 | Publicly available at the NCBI Gene Expression Omnibus (accession no: GSE13539). |
| Guo JU, Ma DK, Mo H, Ball MP, Jang MH, Bonaguidi MA, Balazer JA, Eaves HL, Xie B, Ford E, Zhang K, Ming GL, Gao Y, Song H | 2011 | Neuronal activity modifies the DNA methylation landscape in the adult brain. | https://www.ncbi.nlm.nih.gov/geo/query/acc.cgi?acc=GSE30493 | Publicly available at the NCBI Gene Expression Omnibus (accession no: GSE30493). |
| Halder R, Hennion M, Vidal RO, Shomroni O | 2016 | DNA methylation changes in plasticity genes accompany the formation and maintenance of memory | https://www.ncbi.nlm.nih.gov/geo/query/acc.cgi?acc=GSE74971 | Publicly available at the NCBI Gene Expression Omnibus (accession no: GSE74971). |
| Kim TK, Hemberg M, Gray JM, Costa AM | 2010 | Widespread transcription at neuronal activity-regulated enhancers | https://www.ncbi.nlm.nih.gov/geo/query/acc.cgi?acc=GSE21161 | Publicly available at the NCBI Gene Expression Omnibus (accession no: GSE21161). |
| Lacar B, Linker SB, Jaeger BN, Krishnaswami S, Barron J, Kelder M, Parylak S, Paquola A, Venepally P, Novotny M, O'Connor C, Fitzpatrick C, Erwin J, Hsu JY, Husband D, McConnell MJ, Lasken R, Gage FH | 2016 | Nuclear RNA-seq of single neurons reveals molecular signatures of activation. | https://www.ncbi.nlm.nih.gov/geo/query/acc.cgi?acc=GSE77067 | Publicly available at the NCBI Gene Expression Omnibus (accession no: GSE77067). |
| Lin Y, Bloodgood BL, Hauser JL, Lapan AD, Koon AC, Kim TK, Hu LS, Malik AN, Greenberg ME | 2008 | Activity-dependent regulation of inhibitory synapse development by Npas4. | https://www.ncbi.nlm.nih.gov/geo/query/acc.cgi?acc=GSE11261 | Publicly available at the NCBI Gene Expression Omnibus (accession no: GSE11261). |
| Xiang G, Pan L, Xing W, Zhang L, Huang L, Yu J, Zhang R, Wu J, Cheng J, Zhou Y | 2007 | Identification of activity-dependent gene expression profiles reveals specific subsets of genes induced by different routes of Ca(2+) entry in cultured rat cortical neurons. | https://www.ncbi.nlm.nih.gov/geo/query/acc.cgi?acc=GSE6254 | Publicly available at the NCBI Gene Expression Omnibus (accession no: GSE6254). |

| | | | | |
|---|---|---|---|---|
| Madabhushi R, Gao F, Pfenning AR, Pan L, Yamakawa S, Seo J, Rueda R, Phan TX, Yamakawa H, Pao PC, Stott RT, Gjoneska E, Nott A, Cho S, Kellis M, Tsai LH | 2015 | Activity-Induced DNA Breaks Govern the Expression of Neuronal Early-Response Genes. | https://www.ncbi.nlm.nih.gov/geo/query/acc.cgi?acc=GSE61887 | Publicly available at the NCBI Gene Expression Omnibus (accession no: GSE61887). |
| Samuels BA, Leonardo ED, Dranovsky A, Williams A, Wong E, Nesbitt AM, McCurdy RD, Hen R, Alter M | 2014 | Global state measures of the dentate gyrus gene expression system predict antidepressant-sensitive behaviors. | https://www.ncbi.nlm.nih.gov/geo/query/acc.cgi?acc=GSE43261 | Publicly available at the NCBI Gene Expression Omnibus (accession no: GSE43261). |
| Mehmood T, Schneider A, Sibillec J, Marques-Pereira P, Pannetier S, Ammar M, Dembele D, Thibault-Carpentier C, Rouach N, Hanauer A | 2011 | Transcriptome profile reveals AMPA receptor dysfunction in the hippocampus of the Rsk2-knockout mice, an animal model of Coffin-Lowry syndrome. | https://www.ncbi.nlm.nih.gov/geo/query/acc.cgi?acc=GSE22137 | Publicly available at the NCBI Gene Expression Omnibus (accession no: GSE22137). |
| Ung D, Iacono G, Méziane H, Stunnenberg HG, Vincent JB, Kasri NN, Hérault Y, Laumonnier F | 2017 | Ptchd1 deficiency induces excitatory synaptic and cognitive dysfunctions in mouse | https://www.ncbi.nlm.nih.gov/geo/query/acc.cgi?acc=GSE80312 | Publicly available at the NCBI Gene Expression Omnibus (accession no: GSE80312). |

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
