## [Decision Letter]

Thank you for submitting your article "Forniceal deep brain stimulation induces gene expression and splicing changes that promote neurogenesis and plasticity" for consideration by *eLife*. Your article has been favorably evaluated by a Senior Editor and three reviewers, one of whom is a member of our Board of Reviewing Editors. The following individual involved in review of your submission has agreed to reveal his identity: Sameer Sheth (Reviewer #3).

The reviewers have discussed the reviews with one another and the Reviewing Editor has drafted this decision to help you prepare a revised submission.

Summary:

Deep Brain Stimulation (DBS) has emerged as a promising treatment option in humans for a variety of neuropsychiatric illnesses and forniceal DBS has been suggested in humans and mice to enhance memory formation. In a previous study, the Zoghbi lab used forniceal DBS in a mouse model of Rett Syndrome and found behavioral improvement in tests of hippocampal learning and memory. Here to uncover potential mechanisms of this effect, the authors investigate the gene expression, proteomic, and epigenetic effects of forniceal DBS in a mouse model. They observe upregulation of genes involved in synaptic function and cell survival, as well as normalization of expression of 25% of the genes altered in a loss-of-function mouse model of Rett syndrome. They also found DBS-upregulated genes that are pathologically reduced in other neuropsychiatric disorders, using both mouse data as well as post-mortem tissue from humans with major depression.

The results of this study have significant implications for neuromodulation therapies in general, and DBS in particular. Furthermore this is a massive dataset of gene regulation information following a clinical relevant stimulus that incorporates mouse models of neurological disorders. The data will be broad relevance to a large set of readers.

Essential revisions:

As the authors acknowledge in the manuscript, assessing RNA, protein, and epigenetic changes at one acute time point post-DBS may not be sufficient to provide mechanistic insight for persistent improved behavioral phenotypes upon DBS. Although the information in the manuscript about molecular changes on the minutes time scale is useful from a neurobiological point of view, the authors have a great opportunity to discover the molecular mechanisms by which DBS improves behavior over a long time frame, which is important from a clinical standpoint. For this reason all of three of the reviewers agree that inclusion of at least one more late time point post-DBS, for RNA or protein expression levels, or an attempt to address a causal relationship between any of the identified gene expression or DNA methylation changes, would make a big impact. Unless the authors feel this is prohibitively difficult, we strongly encourage the authors to include these data in a revision.

---

## [Author Response]

Essential revisions:As the authors acknowledge in the manuscript, assessing RNA, protein, and epigenetic changes at one acute time point post-DBS may not be sufficient to provide mechanistic insight for persistent improved behavioral phenotypes upon DBS. Although the information in the manuscript about molecular changes on the minutes time scale is useful from a neurobiological point of view, the authors have a great opportunity to discover the molecular mechanisms by which DBS improves behavior over a long time frame, which is important from a clinical standpoint. For this reason all of three of the reviewers agree that inclusion of at least one more late time point post-DBS, for RNA or protein expression levels, or an attempt to address a causal relationship between any of the identified gene expression or DNA methylation changes, would make a big impact. Unless the authors feel this is prohibitively difficult, we strongly encourage the authors to include these data in a revision.

To address the reviewers’ question about the molecular mechanisms of DBS at a later time point, we performed chronic DBS on a new cohort of mice to evaluate gene expression using RNA-sequencing. There were 15 samples: 7 MeCP2-heterozygous females (3 sham, 4 DBS) and 8 WT females (4 sham, 4 DBS). The mice underwent two weeks of DBS (1hr per day for 14 days), then recovered for 17 days before having their dentate gyri collected. This time point for tissue collection was chosen based on the Hao et al. paper as this was the earliest time at which behavior was assessed following chronic DBS.

The results of this sequencing experiment are included in the manuscript as the new Figure 6.

We find that the response of WT and Het female mice to acute DBS is similar to that of male mice, and, importantly, there is a trend towards improved expression of misregulated genes in *Mecp2*-heterozygous female samples that received chronic DBS. Thus, one way in which chronic DBS likely improves learning and memory in Rett mice is through rescued gene expression. The chronic DBS RNA-Seq data can be found on GEO at the following link: https://www.ncbi.nlm.nih.gov/geo/query/acc.cgi?acc=GSE111703